# A combination of chitooligosaccharide and lipochitooligosaccharide recognition promotes arbuscular mycorrhizal associations in *Medicago truncatula*

Feng Feng [1], Jongho Sun[1], Guru V. Radhakrishnan [2], Tak Lee [1], Zoltán Bozsóki[3], Sébastien Fort [4], Aleksander Gavrin [1], Kira Gysel [3], Mikkel B. Thygesen [5], Kasper Røjkjær Andersen [3], Simona Radutoiu[3], Jens Stougaard [3] & Giles E.D. Oldroyd [1]*

Plants associate with beneficial arbuscular mycorrhizal fungi facilitating nutrient acquisition. Arbuscular mycorrhizal fungi produce chitooligosaccharides (COs) and lipo-chitooligosaccharides (LCOs), that promote symbiosis signalling with resultant oscillations in nuclear-associated calcium. The activation of symbiosis signalling must be balanced with activation of immunity signalling, which in fungal interactions is promoted by COs resulting from the chitinaceous fungal cell wall. Here we demonstrate that COs ranging from CO4-CO8 can induce symbiosis signalling in *Medicago truncatula*. CO perception is a function of the receptor-like kinases *Mt*CERK1 and LYR4, that activate both immunity and symbiosis signalling. A combination of LCOs and COs act synergistically to enhance symbiosis signalling and suppress immunity signalling and receptors involved in both CO and LCO perception are necessary for mycorrhizal establishment. We conclude that LCOs, when present in a mix with COs, drive a symbiotic outcome and this mix of signals is essential for arbuscular mycorrhizal establishment.

[1] Sainsbury Laboratory, University of Cambridge, 47 Bateman Street, Cambridge CB2 1LR, UK. [2] Department of Cell and Developmental Biology, John Innes Centre, Norwich NR4 7UH, UK. [3] Department of Molecular Biology and Genetics, Aarhus University, Aarhus 8000 C, Denmark. [4] Université de Grenoble Alpes, CNRS, CERMAV, 38000 Grenoble, France. [5] Department of Chemistry, University of Copenhagen, Frederiksberg 1871 C, Denmark. *email: gedo2@cam.ac.uk

Plants are exposed to an array of microorganisms that include potential mutualists and pathogens and must monitor these interactions to coordinate appropriate responses, whether restriction of pathogens or promotion of mutualists. This decision making is primarily regulated by two signaling pathways in the plant: symbiosis signaling that promotes microbial associations and immunity signaling that restricts them[1–4]. These pathways are predominantly associated with the regulation of intercellular and intracellular colonization of plant tissues, however, it is becoming clear that these pathways also impact on the microbiomes that form in association with plants[5,6].

Primary recognition of fungi involves perception of chitinaceous molecules derived from the fungal cell wall. N-acetyl chitooctaose (CO8) is particularly active in promoting immunity signalling[2,7,8], although other N-acetyl chitooligosaccharides (COs), including shorter chain COs such as N-acetyl chitotetraose (CO4), can activate immunity signalling, albeit at lower efficiencies[7,8]. Recognition of CO8 involves LysM-containing receptor-like kinases with the capability to bind directly to chitinaceous molecules through the LysM domains[9–16]. In both rice and Arabidopsis thaliana the perception of CO8 involves two LysM-containing receptors, one possessing an active intracellular kinase domain (CERK1[12,14]) and one containing either an inactive kinase domain (LYK5 of A. thaliana[9]) or a truncated intracellular domain (CEBiP in rice[11]). A similar situation has been demonstrated to exist in legumes, with Medicago truncatula requiring the AtCERK1 homolog LYK9 (renamed here MtCERK1) and the AtLYK5 homolog LYR4 for CO8 activation of immunity signalling and appropriate resistance to fungal pathogens[7]. Perception of chitin by these receptors leads to the activation of plant defenses through production of reactive oxygen species (ROS), promotion of MAP kinases and activation of a calcium influx across the plasma membrane[2,7,9].

Chitinaceous molecules also control beneficial fungal associations, with arbuscular mycorrhizal fungi producing both COs and lipochitooligosaccharides (LCOs)[17,18], which possess an N-acyl moiety on the terminal non-reducing sugar of the chitin chain[19]. Both LCOs and short chain COs (CO4 and CO5) can promote symbiosis signalling in M. truncatula with resultant oscillations in nuclear calcium levels[17,20]. Short chain COs activate symbiotic calcium oscillations in a range of species (M. truncatula, Lotus japonicus, carrot and rice[17,20]) and in rice this is a function of the CO receptor CERK1[21], that is necessary for appropriate arbuscular mycorrhizal colonization[22–25]. Arbuscular mycorrhizal fungi and rhizobial bacteria produce LCOs and the receptor complex in legumes involved in this perception contains two LysM-containing receptor-like kinases: LYK3 and NFP in M. truncatula (NFR1 and NFR5 in L. japonicus)[26–33]. Mutations in these receptors abolish the interaction with nitrogen-fixing rhizobial bacteria, highlighting the essential role LCO perception plays in this bacterial symbiosis. In contrast, plants mutated in the LCO receptors have wild type or only marginally reduced colonization by arbuscular mycorrhizal fungi[23,34].

Beneficial symbiotic microorganisms must evade plant defenses in order to colonize the root of their host plant. This may in part be regulated by effector molecules delivered into the cell from the symbiotic fungi or bacteria[35]. However, it has also been demonstrated that the plant actively suppresses immunity and this occurs following recognition of both CO4 and LCOs in A. thaliana and other plants[36]. In this work, we have used a combination of cell biology and genetics to characterize the relative contributions of COs and LCOs for establishment of arbuscular mycorrhizal associations in M. truncatula. We show that COs cannot be separated into those that activate symbiosis versus immunity responses, rather it appears that COs equally elicit symbiotic and immunity signalling, with longer chain COs, such as CO8, showing greater activity. Receptors responsible for both CO and LCO recognition are required in the establishment of arbuscular mycorrhizal associations and we show how the recognition of mixes of COs and LCOs emphasizes a symbiotic, over immunogenic, response.

## Results

**CO8 activates symbiotic calcium signalling.** Symbiosis signalling involves the promotion of periodic calcium oscillations restricted to the nuclear region[37–39]. M. truncatula shows symbiotic calcium oscillations following treatment with either CO4 or LCOs[17,20], but not to the immunity elicitor flg22 (0/18 epidermal cells showed calcium responses following treatment of $10^{-5}$ M flg22). We found that nuclear-associated calcium oscillations were activated following treatments with CO8 (Fig. 1a, b), previously thought to function primarily as an immunity signal[2,40]. To test the degree to which other CO molecules activate symbiosis signalling we assessed the induction of nuclear calcium oscillations by all CO molecules between CO2 and CO8. CO4, CO5, CO6, CO7 and CO8 all activate nuclear calcium oscillations, with comparable activities when applied at $10^{-8}$ M. However, neither CO2 nor CO3 could activate nuclear calcium oscillations (Supplementary Table 1).

The CO8-induced nuclear calcium responses of roots in whole plants of M. truncatula show a periodicity similar in nature to those induced by CO4 (Fig. 1a). Dose response curves that assess the number of cells responding with nuclear calcium oscillations across a range of elicitor concentrations, indicate that CO8 is more active in M. truncatula roots than CO4 (Fig. 1c). CO4 can induce immunity signalling in M. truncatula[7], but the degree of the CO4 response is much less than CO8 and this is comparable to the reduced activity of CO4 in promotion of symbiosis signalling. In contrast, SmLCOs are much more active than either CO4 or CO8 for induction of nuclear calcium oscillations (Fig. 1c), but show no induction of ROS or MAPKs, markers of immunity signalling[7]. The LCO produced by the mycorrhizal fungus Rhizophagus irregularis (NS-LCO) shows an activity within a similar range as CO4/CO8, but is slightly less active than either molecule (Fig. 1c). The concentrations of CO8 required for the induction of symbiosis signalling are comparable to those required for induction of immunity signalling[7], implying that the receptors involved in CO8 perception for immunity or symbiosis signalling must have comparable activation kinetics.

It was previously assumed that CO8 only functions as an immunity elicitor and therefore, it was very surprising to see CO8 induction of symbiosis signalling. To validate that this response was indeed a function of CO8 we first tested the purity of our CO8 samples and found that they were not contaminated with either CO4 or CO5 (Supplementary Fig. 1a). Plant roots exude a number of chitinases and it is possible that treating roots with CO8 leads to an accumulation of shorter chain COs as degradation products of CO8 and the resultant short chain COs could then activate symbiosis signalling. CO8 treated on M. truncatula roots does indeed get broken down quickly, with a 50% reduction in total CO8 levels after 10 min incubation on M. truncatula roots (Supplementary Fig. 1b). The chitinase inhibitor acetazolamide[41] reduces the rate and degree of this CO8 degradation (Supplementary Fig. 1b) and consistent with this we found that co-treatment of acetazolamide with CO8 enhanced the activation of immunity signalling as measured by the activation of ROS (Supplementary Fig. 2a). The symbiotic response to CO8 was also enhanced by co-treatment with acetazolamide: we found a significantly reduced activation time for CO8 promotion of nuclear calcium oscillations

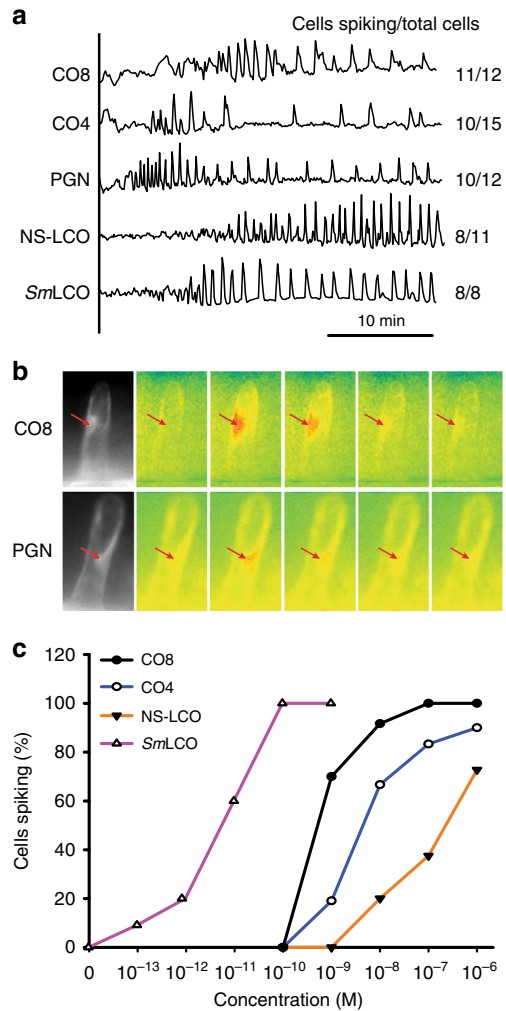

**Fig. 1** COs and LCOs activate symbiotic calcium oscillations. **a** Representative traces of $10^{-8}$ M CO8, $10^{-8}$ M CO4, 0.8 mg/ml PGN, $10^{-8}$ M NS-LCO and $10^{-9}$ M SmLCO-induced calcium oscillations in *M. truncatula* trichoblasts on lateral roots. The *y*-axis represents the ratio of YFP to CFP in arbitrary units. Numbers denote responsive cells relative to total cells analysed. **b** CO8 and PGN-induced calcium oscillations were restricted to the nuclear region (arrows indicate the location of the nucleus). Images were taken every five seconds covering a single calcium transient. **c** Dose response curves for CO and LCO induction of calcium oscillations in *M. truncatula* lateral root trichoblasts. SmLCO: sulphated LCO produced by *S. meliloti*. NS-LCO: non-sulphated LCO produced by *R. irregularis*

(Supplementary Fig. 2b) and a greater sensitivity of *M. truncatula* root cells to CO8 in the presence of acetazolamide (Supplementary Fig. 2c). Acetazolamide alone cannot induce nuclear calcium oscillations (Supplementary Table 2). We conclude that CO8 itself most likely acts as the elicitor, but breakdown products of CO8 can also act as elicitors, provided they are larger than CO3.

Arbuscular mycorrhizal fungi produce LCOs as well as short chain COs[17,18], however, they have not yet been reported to produce CO8. However, considering that CO8 is a byproduct of chitinaceous fungal cell walls, we would expect that mycorrhizal fungi, like their pathogenic fungal counterparts, will produce CO8. Mass spectral analysis of germinated spore exudates from *R. irregularis* were too complex in the region of CO8 to definitively state whether CO8 was present or not. Hence, we

instead took an indirect approach to assess for the presence of CO8 in *R. irregularis* exudates. We treated *Arabidopsis* leaf discs with *R. irregularis* germinated spore exudates and observed a rapid ROS burst indicative of the activation of immunity signalling. This ROS response was greatly reduced in *Atcerk1* mutants revealing that the chitinaceous elicitor of *AtCERK1* must be present in these exudates of *R. irregularis* (Supplementary Fig. 3a). We conclude that chitinaceous elicitors are produced by arbuscular mycorrhizal fungi and they promote both symbiosis and immunity signalling.

**MtCERK1 and LYR4 are receptors for CO8.** To validate that these calcium responses were a function of symbiosis signalling we assessed CO8-induced calcium responses in plants mutated in components of the symbiosis signalling pathway necessary for the establishment of nuclear calcium oscillations: *DMI1* and *DMI2*[42–44]. Both *dmi1* and *dmi2* mutants abolished CO8-induced nuclear calcium oscillations (Fig. 2a), indicating a function for symbiosis signalling in this CO8-induced calcium response. *DMI2* encodes a receptor-like kinase able to associate with LCO receptors in *L. japonicus*[45,46]. Interestingly, while this receptor is essential for both CO and LCO induction of symbiotic calcium oscillations, it is not required for CO8-induced defence responses (Supplementary Fig. 4). Mutations in *dmi1* and *dmi2* abolish symbiotic signalling and beneficial microbial associations[17,20,42–44], but have no effect on immunity signalling and little or no effect on pathogen interactions[47–50]. We therefore conclude that CO8-induced nuclear calcium oscillations are a function of symbiosis signalling.

The LCO receptors *NFP* and *LYK3* were not required for CO8 induction of calcium oscillations (Fig. 2a and Supplementary Fig. 5a), implying the involvement of an alternative receptor complex. *M. truncatula LYR4* and *MtCERK1* act as CO8 receptors for activation of immunity signalling and are necessary for restriction of the fungal pathogen *Botrytis cinerea*[7], as well as immunity responses to *R. irregularis* germinated spore exudates (Supplementary Fig. 3b). To assess if these CO8 receptors were also necessary for the activation of symbiotic responses we crossed these receptor mutants into the calcium reporter Yellow cameleon YC3.6 to measure the activation of symbiotic calcium responses. We found that *MtCERK1* was essential for CO8-induced symbiotic calcium oscillations in both trichoblasts (epidermal root hair cells) and atrichoblasts (epidermal root cells lacking root hairs; Fig. 2a and Supplementary Fig. 5a). Surprisingly, *LYR4* was not required for CO8-induced calcium responses in atrichoblasts, but was required for CO8-induced calcium oscillations in trichoblasts (Fig. 2a and Supplementary Fig. 5a), suggesting additional receptors in atrichoblasts that may compensate for the absence of *LYR4*. Complementation of *Mtcerk1* and *lyr4* with their respective wild-type genes reinstated CO8 activation of symbiotic calcium oscillations (Supplementary Table 3). We have previously demonstrated that *MtCERK1* and *LYR4* are required for activation of immunity signalling in *M. truncatula* by both CO4 and CO8[7]. In a similar fashion, we found that *MtCERK1* was essential for CO4-induced calcium oscillations in both trichoblasts and atrichoblasts, while *LYR4* was required for CO4-induced calcium oscillations in trichoblasts, but not in atrichoblasts (Supplementary Fig. 5b, c). We conclude that *MtCERK1* and *LYR4* are responsible for CO8 and CO4 activation of both immunity and symbiosis signalling.

Our genetic evidence implies a function for *MtCERK1* and *LYR4* in the perception of both CO4 and CO8. To validate these observations we generated beads fused to either CO4 or CO8, using biotinylated versions of CO4 and CO8, attached to Streptavidin beads[51]. *MtCERK1* showed strong binding to CO8

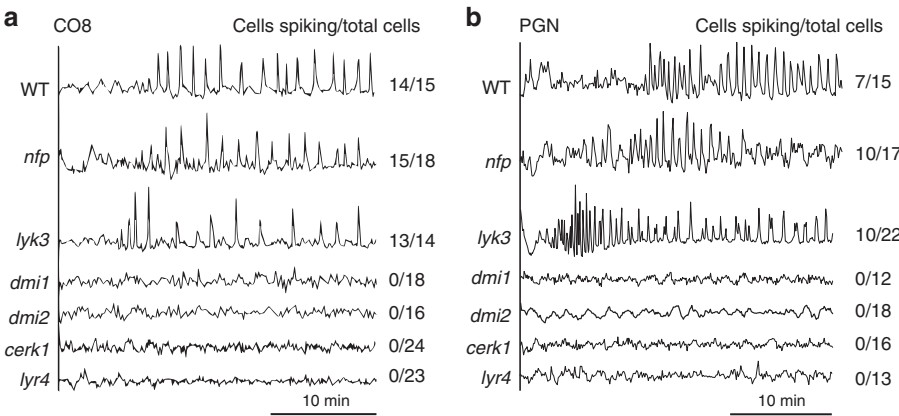

**Fig. 2** CO8 and PGN activate calcium oscillations dependent on *DMI1, DMI2, MtCERK1* and *LYR4*. Representative calcium traces of *M. truncatula* trichoblasts of lateral roots responding to $10^{-8}$ M CO8 (**a**) and 0.8 mg/ml PGN (**b**) in wild type and different mutants. The traces denote the ratio of YFP to CFP in arbitrary units. Numbers indicate cells responding compared to total cells analyzed

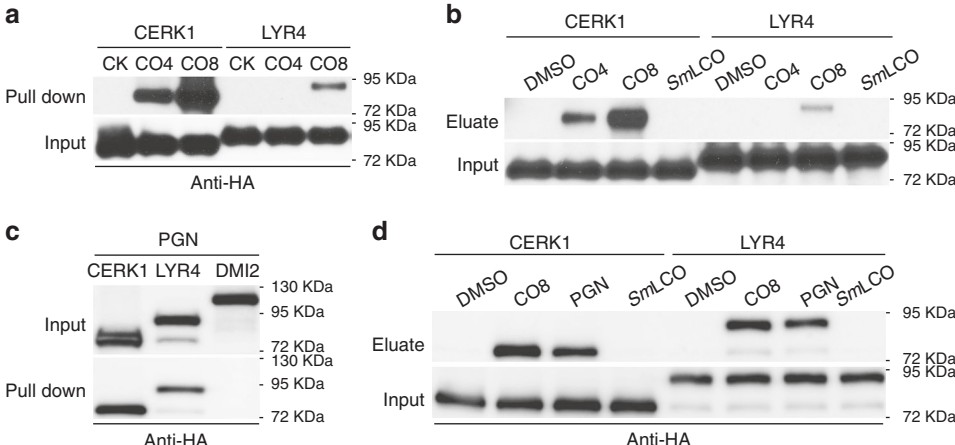

**Fig. 3** LYR4 and *Mt*CERK1 bind to COs and PGN. **a** LYR4 and *Mt*CERK1 binding with streptavidin beads fused to either CO4 or CO8. CK: streptavidin beads alone. **b** *Mt*CERK1 binding with CO4 beads can be eluted by addition of CO4 and CO8. LYR4 binding with CO8 beads can be eluted with CO8, but not CO4. **c** LYR4 and *Mt*CERK1, but not DMI2, bind to insoluble PGN and **d** bound proteins can by eluted by CO8 and the soluble fraction of PGN, but not *Sm*LCO. The protein expression level of LYR4 and *Mt*CERK1 used in the binding assay is shown as input

and weaker binding to CO4, while LYR4 showed some binding to CO8, but no binding to CO4 (Fig. 3a). The *Mt*CERK1 binding to CO4 beads could be eluted with addition of either CO4 or CO8, but not *Sm*LCO (Fig. 3b). However, CO8 was more effective than CO4 at eluting *Mt*CERK1 from the CO4 beads. The LYR4 binding to CO8 could only be competed with addition of CO8 (Fig. 3b). From this work, we conclude that *Mt*CERK1 can bind both CO4 and CO8, but has a preference for CO8, while LYR4 can bind CO8, but it appears with less efficiency than *Mt*CERK1.

**CO8 activates symbiotically relevant gene expression**. To further explore the relative overlap between immunity and symbiosis signalling, we explored the transcriptional profiles of roots treated with either *Sm*LCO, CO4, CO8 or flg22 (Supplementary Data 1). *R. irregularis* produces a sulphated LCO very similar to the predominant LCOs produced by *S. meliloti*, that also shows similar elicitation activity as *Sm*LCO for induction of symbiotic calcium oscillations in *M. truncatula*[18,20]. In this study we used *Sm*LCO as a measure of the LCO-induced response. Of the genes that were significantly upregulated by greater than twofold we observed a large overlap between CO8 and flg22 (54% of CO8-induced genes were also induced by flg22; Supplementary

Fig. 6a), with much less overlap between *Sm*LCO and flg22 (25% of *Sm*LCO-induced genes were upregulated in flg22 treatments; Supplementary Fig. 6a). By contrast, 51% of genes induced by *Sm*LCO were also activated by CO8 (Supplementary Fig. 6a), revealing a sizable overlap between the CO8 and *Sm*LCO responses.

Flg22-induced genes show induction by COs, less induction by *Sm*LCO and very little induction by mycorrhizal fungi (Fig. 4a). We conclude that these flg22-induced genes most likely represent the immunity response and this is supported by the presence of a number of genes known to function in plant defenses among this group (Supplementary Fig. 7). A second group of genes are induced by CO8, *Sm*LCO and mycorrhizal colonization (Fig. 4a), but not flg22. These include a number that are known to function during mycorrhizal colonization, for instance *Vapyrin, HA1, D27, PUB1*[52] and a symbiotic subtilase[53] (Supplementary Fig. 7). To assess the dependency of these gene expression changes on the CO and LCO receptors, we analyzed responses of CO4, CO8 and *Sm*LCO in *Mtcerk1, lyr4* and *nfp* mutants. For this work, we generated a new *nfp* mutant allele in the *M. truncatula* ecotype R108 (Supplementary Fig. 8). This *nfp* allele allowed ease of comparison with *Mtcerk1* and *lyr4*, that are also in the R108 ecotype. 86.43% of the genes that were activated by CO8 were

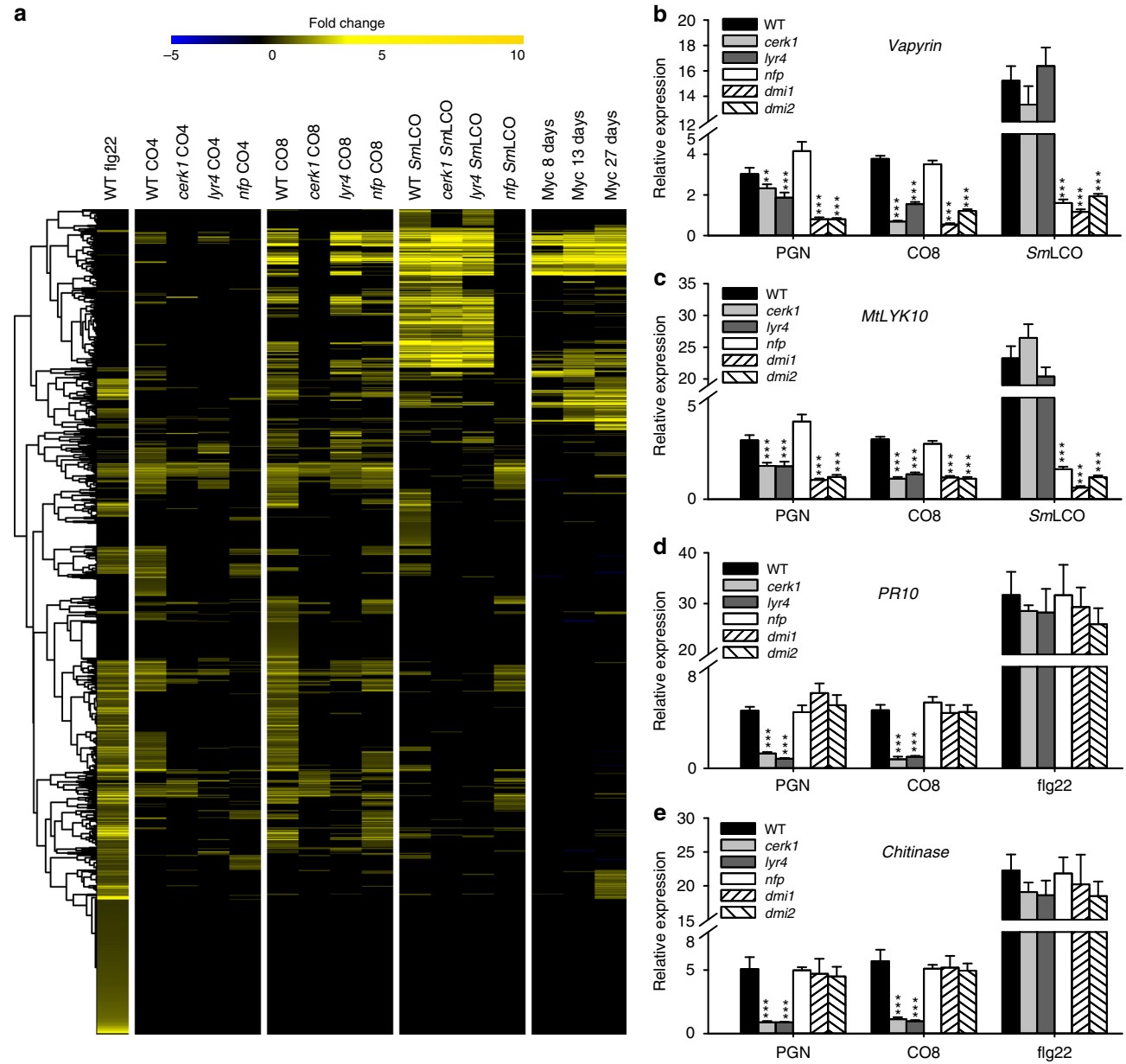

**Fig. 4** *MtCERK1* and *LYR4* are required for CO8 and PGN-induced gene expression. **a** A heat map of genes induced by flg22, CO4, CO8 and *Sm*LCO (6 h treatment). The expression of these genes in mutants and during mycorrhizal colonization is shown. qRT-PCR validation of representative symbiotic genes (**b** and **c**) and defense genes (**d** and **e**) induced by PGN, CO8, *Sm*LCO and flg22. *M. truncatula* wild type and mutant roots were treated with 0.4 mg/ml PGN, $10^{-7}$ M CO8, $10^{-8}$ M *Sm*LCO and $10^{-7}$ M flg22 using comparable conditions to the material used in the RNA-seq analysis. This experiment was repeated three times with similar results (mean ± s.e.m.; $n = 3$; significant difference relative to wild type by Student's $t$-test, ***$P < 0.001$; **$P < 0.05$)

dependent on *MtCERK1*, while 65.62% of these genes showed *LYR4* dependence (Fig. 4a). The reduced dependency of the CO8-response on *LYR4*, as oppose to *MtCERK1*, is consistent with the observation that CO8-induced calcium oscillations are maintained in atrichoblasts (Supplementary Fig. 5a). Surprisingly, there was also a function for *NFP* in the CO8 response, with 47.95% of the CO8-induced genes being *NFP*-dependent. The degree of transcriptional response to CO4 was greatly attenuated compared to CO8, consistent with CO8 being a more active elicitor. However, there are some genes induced specifically by CO4, implying a degree of specificity in the CO4 response (Fig. 4a). The CO4 transcriptome was highly dependent on both *MtCERK1* (80.16% of genes) and *NFP* (86.96% of genes), with less dependency on *LYR4* (72.27% of genes). The majority (93.42%) of

genes induced by *Sm*LCO were dependent on *NFP*. *MtCERK1* and *LYR4* also contribute to the *Sm*LCO response but by a lower degree, 46.03% and 47.17% respectively. While these analyses, along with previous work, support a primary function for *MtCERK1* and *LYR4* in the CO response[7] and *NFP* in the LCO response[20,54], they also reveal contributions from all receptors in CO and LCO perception.

We validated these gene inductions using quantitative reverse transcription polymerase chain reaction (qRT-PCR) of *Vapyrin*, *MtLYK10*[55], *HA1* and *D27* induction. Included in this analysis were the *Mtcerk1*, *lyr4* and *nfp* receptor mutants, but also mutants in the symbiosis signalling components *dmi1* and *dmi2*. We observed CO8 and *Sm*LCO induction of *Vapyrin*, *MtLYK10*, *HA1* and *D27* using qRT-PCR, and in all cases this induction was

dependent on *DMI1* and *DMI2* (Fig. 4b, c and Supplementary Fig. 6b, c), revealing a function for symbiosis signalling in these gene expression changes. All genes induced by CO8 were *MtCERK1* and *LYR4* dependent and in a few cases *NFP* dependent, while *Sm*LCO induction of these genes was *NFP* dependent, but *MtCERK1/LYR4* independent (Fig. 4b, c and Supplementary Fig. 6b, c). Interestingly, the defense-related genes induced by CO8 were dependent on *MtCERK1* and *LYR4*, but independent of *DMI1* and *DMI2* (Fig. 4d, e and Supplementary Fig. 6d, e). *DMI2* is not required for plant immunity (Supplementary Fig. 4), suggesting that the differential activation of immunity or symbiosis signalling by *Mt*CERK1/LYR4 might be explained by the presence of DMI2 as a co-receptor. Our work reveals that CO8 activates genes associated with both immunity and symbiosis and this correlates with its ability to activate both immunity and symbiosis signalling.

**Recognition of COs with LCOs promotes symbiosis over immunity.** *R. irregularis* appears to produce both COs and LCOs[17,18] and we would expect that a combination of these signals will be presented to root cells as the fungus approaches. It has already been demonstrated that LCOs can inhibit the degree of immunity signalling induced by CO8 or flg22[36]. Here we further explored the effect of these signals when presented in combination. Consistent with previous work[36], when *Sm*LCO was combined with CO8 we found a reduction in the resultant ROS and MAPK responses (Fig. 5a, b), within 15 min of treatment with *Sm*LCO (Fig. 5a). qRT-PCR analysis of two genes induced by flg22 and CO8 (Fig. 5c), a *PR10* and a chitinase, revealed that co-treatment with *Sm*LCO inhibited CO8 induction of these genes (Fig. 5c), in disagreement with a recent report[56]. The combination of LCOs and COs inhibits defense-related gene expression, but these two signals act synergistically to enhance the degree of symbiotic gene expression, as measured using qRT-PCR of *Vapyrin* and *HA1* (Fig. 5c).

To demonstrate whether LCO suppression of plant immunity is a function of the symbiosis signalling pathway, we assessed *Sm*LCO modification of plant immunity in mutants defective in symbiosis signalling. Only *NFP*, but not other genes in this pathway, were necessary for *Sm*LCO suppression of immunity (Fig. 6a), suggesting an independent signalling function downstream of *NFP*, that is different to the symbiosis signalling pathway. Overexpression of *NFP* alone cannot inhibit *MtCERK1*-induced cell death (Supplementary Fig. 9) and treatment with the protein synthesis inhibitor cycloheximide blocked the *Sm*LCO suppression of immunity signalling (Supplementary Fig. 10a), pointing at novel protein synthesis being involved. LCO treatment enhances root susceptibility to the pathogenic oomycete *Phytophthora palmivora*, and this was dependent on *NFP*, but not other components of symbiosis signalling (Fig. 6b, c). Consistent with a previous report[49,57], *nfp* showed increased susceptibility to *P. palmivora* (Fig. 6b, c) and in addition we found that mutants of *Mtcerk1* and *lyr4* showed greater susceptibility (Supplementary Fig. 11), implying roles for these receptors in pathogenic root associations.

Our work indicates that when presented together LCOs alter the nature of the CO response towards a symbiotic outcome, with reduced immunity and enhanced symbiosis signalling. This suggests that such a combination of signal perception will be important during the interaction with mycorrhizal fungi. *NFP* is necessary for all LCO responses[20,54], whether of fungal or bacterial origin, implying that this receptor may have functions beyond simply a role in rhizobial interactions, as already demonstrated[26,27]. To test the importance of the mix of COs and LCOs we generated a number of double mutant

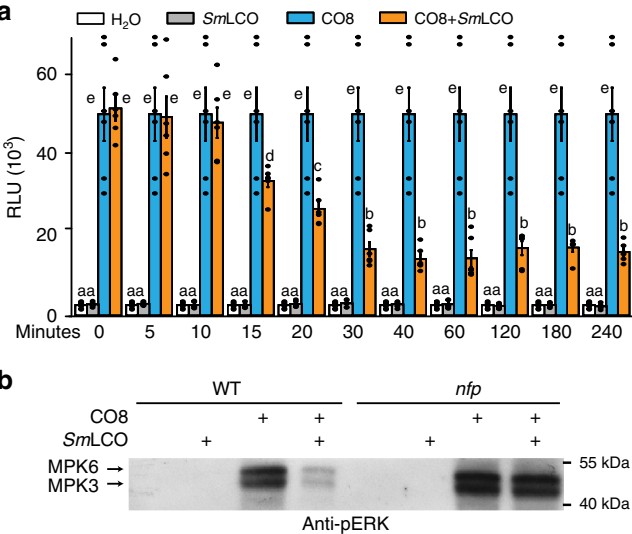

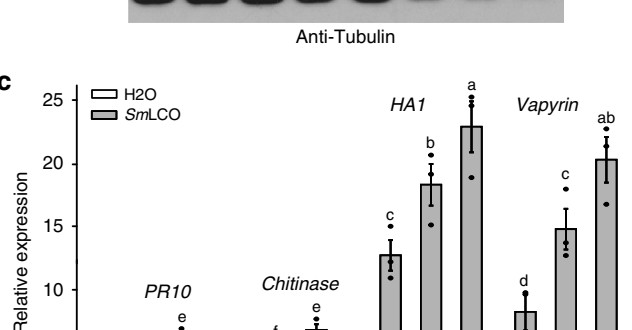

**Fig. 5** A combination of CO8 and *Sm*LCO can promote symbiosis and inhibit immunity signalling. **a** *M. truncatula* wild-type roots were pretreated with $10^{-8}$ M *Sm*LCO for different time points (minutes) before incubation with $10^{-6}$ M CO8 for induction of ROS. The significant groupings were calculated with a Mann–Whitney Rank Sum Test (mean ± s.e.m., $n = 6$; $P < 0.05$). This experiment was repeated twice with similar results. **b** MAPK activation in wild type and *nfp* roots induced by $10^{-6}$ M CO8 with or without 30 min pretreatment of $10^{-8}$ M *Sm*LCO. **c** Equivalent treatment as **b** for qRT-PCR detection of gene expression testing immunity reporters *PR10* and a *Chitinase* and symbiosis reporters *HA1* and *Vapyrin* in response to CO8 and PGN, with or without *Sm*LCO. Relative fold change compared to individual water treatments are shown. Letters denote statistically significant groupings calculated with Mann–Whitney Rank Sum Test (mean ± s.e.m., $n = 3$; $P < 0.05$), this experiment was repeated twice with similar results

combinations that combined mutations in the LCO receptors *NFP* and *LYK3* and the CO receptors *MtCERK1* and *LYR4*. Because *LYK3* and *LYR4* are in close proximity on the *M. truncatula* genome it was not possible to generate the *lyk3/lyr4* double mutant. All lines that carried a mutation in *MtCERK1* showed significant reductions in mycorrhizal fungal colonization at both three and five weeks post inoculation with *R. irregularis* spores, in support of recent work[24]. In contrast, mutation of *LYR4* alone appeared to have no effect (Fig. 7a). This differential dependence on the CO8 receptors likely reflects the cell specific nature of *lyr4*, since mycorrhizal fungi preferentially colonize the

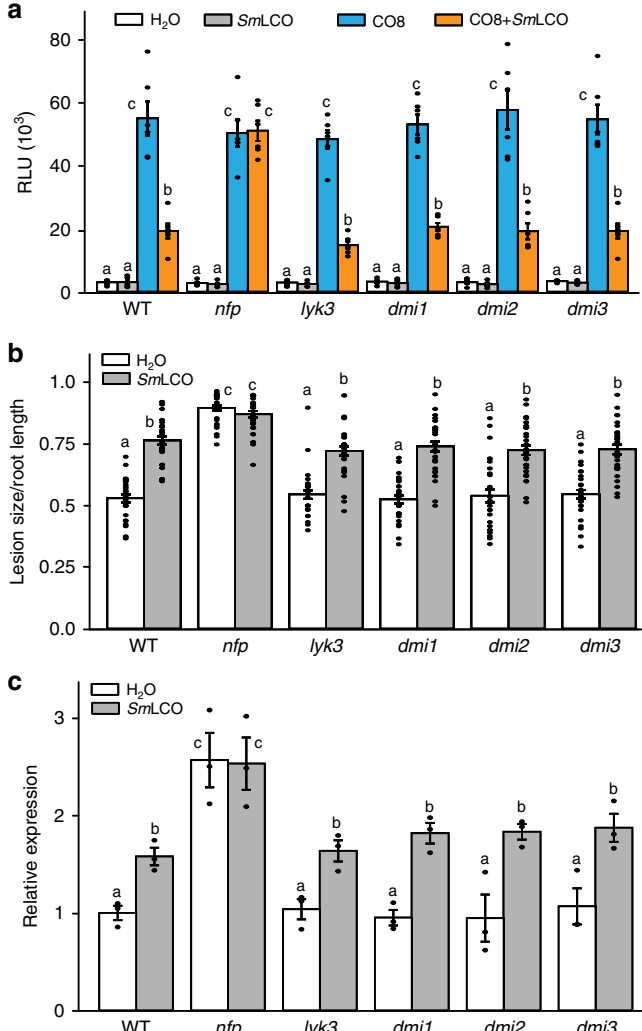

**Fig. 6** *NFP* but not other symbiosis signalling components are required for LCO suppression of plant immunity. **a** *M. truncatula* wild-type and mutant roots were pre-treated with $10^{-8}$ M *Sm*LCO for 30 min before challenging with CO8 for ROS induction. This experiment was repeated three times with similar results. **b** *M. truncatula* roots with or without *Sm*LCO treatments were inoculated with *P. palmivora* and the lesion size quantified relative to root length or **c** quantity of pathogen measured using amplification of *P. palmivora EF1a* relative to *M. truncatula H2A*. These experiments were repeated twice with similar results. Letters denote statistically significant groupings calculated with Mann-Whitney Rank Sum Test (mean ± s.e.m., $n = 6$ for **a**, $n = 30$ for **b** and $n = 3$ for **c**, $P < 0.05$)

root through atrichoblasts, that show normal CO-induced calcium responses in *lyr4* (Supplementary Fig. 5). With the high concentrations of fungal spores used in the inoculation, we observed no phenotype in the *lyk3* mutant and no phenotypes in the *nfp* mutant (Fig. 7a). However, when *nfp* was combined with the *Mtcerk1* mutation, we observed a severe reduction in fungal colonization (Fig. 7a), that was not present in *Mtcerk1/lyk3* double mutants. All fungal structures were reduced in *Mtcerk1* and *Mtcerk1/nfp* (Fig. 7b), although arbuscules that did develop appeared normal (Fig. 7c). We conclude that both CO and LCO perception contributes to mycorrhizal colonization, but additional receptors must also function in this symbiosis, since some fungal colonization still occurs in *Mtcerk1/nfp* (there remain 21 functional LysM-RLKs in this double mutant).

**Peptidoglycan is also a dual immunity/symbiotic elicitor**. Like chitin, PGN that is present in bacterial envelopes, possesses *N*-acetyl glucosamine residues, but additional sugars are present that are linked through a combination of bonds to create a much more complex network of sugar moieties than chitin[58]. PGN perception in *Arabidopsis* and rice involves *CERK1*[59,60] and therefore we explored whether PGN may mirror CO8 in acting as a dual immunity and symbiosis elicitor. PGN activates nuclear calcium oscillations in a manner comparable to CO8 (Fig. 1a, b) and this is dependent on the symbiosis signalling components *DMI1* and *DMI2*, as well as the receptors *MtCERK1* and *LYR4* (Fig. 2b and Supplementary Fig. 5d). Like CO8, we observed some calcium oscillations in atrichoblasts of *lyr4*, as well as *Mtcerk1*, but in the case of PGN, there still appeared to be a reduction in the response of atrichoblasts in these mutants (Supplementary Fig. 5d, e). We found that both *Mt*CERK1 and LYR4 could bind to PGN (Fig. 3c) and this binding could be eluted with addition of either CO8 or PGN (Fig. 3d). Like CO8, PGN acts as an immunity elicitor in *M. truncatula* in a manner dependent on *MtCERK1* and *LYR4* (Supplementary Fig. 12) and this induction of immunity signalling could be suppressed by addition of LCOs (Supplementary Fig. 10b, c). We conclude that PGN appears to function in a similar fashion as CO8, acting as both an immunity and a symbiotic elicitor, however, unlike mycorrhization, we were unable to observe any phenotypes during rhizobial colonization in the *Mtcerk1* or *lyr4* mutants[7].

## Discussion

Symbiosis signalling is preserved across diverse plant species and is necessary for perception of arbuscular mycorrhizal fungi and their accommodation in plant roots[61]. This signalling pathway has also been recruited during the evolution of nitrogen-fixing symbioses in legumes and actinorhizal species[62,63], that use this pathway to perceive LCOs produced by rhizobial bacteria[61] or an unknown signalling molecule produced by Frankia bacteria[63]. The presence of the genetic components that make up symbiosis signalling correlate very closely with the capability to enter symbiotic associations: plant species that have lost arbuscular mycorrhizal associations, lose the symbiosis signalling pathway[64], suggesting the signalling pathway is redundant in the absence of beneficial microbial associations. The role of symbiosis signalling during symbiotic associations is absolute: mutations in this pathway completely abolish colonisation of arbuscular mycorrhizal fungi and rhizobial bacteria. No such major role has yet been found for this pathway during pathogen or parasitic associations[47–49], instead symbiosis genes or their homologs appear to play only marginal roles in colonisation by a very limited set of pathogens or parasites[65,66]. We therefore conclude that symbiosis signalling functions primarily in the establishment of mutualistic associations and despite the presence of this signalling pathway over hundreds of millions of years of co-evolution, there is very little evidence that pathogens or parasites have utilised this signalling pathway to promote their own colonisation.

The receptor *NFP* is essential for all LCO responses in *M. truncatula*[20] and required for associations with rhizobial bacteria[27]. However, the effect of *NFP* is broader, since it contributes to resistance against both oomycete and fungal pathogens[49,50,57]. Here we demonstrate that *NFP* also contributes to colonization by arbuscular mycorrhizal fungi, but this effect can only be observed when combined with the *Mtcerk1* mutant. In addition to the role of *NFP* in LCO responses[20], we have observed some dependence on *NFP* in the CO response. This is particularly apparent in the CO4 transcriptional response and the CO4 induction of calcium oscillations[20], but we also observed some dependence on *NFP* in the CO8 transcriptional response. Hence, *NFP* may be involved in

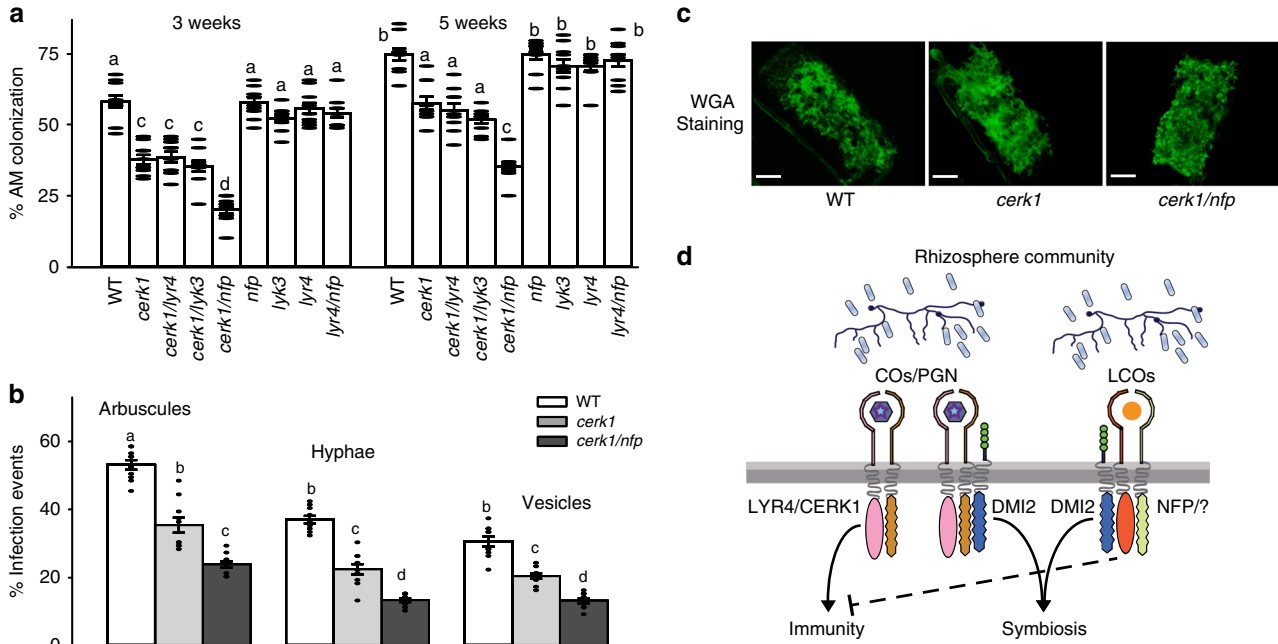

**Fig. 7** *MtCERK1* and *NFP* are required for arbuscular mycorrhizal colonization. **a** *R. irregularis* colonization shown as percent root length colonization assessed at 3 and 5 weeks post inoculation. **b** Individual fungal structures were quantified at 3 weeks post inoculation. This experiment was repeated three times with similar results. **c** Representative images of arbuscules in wild type and mutant plants. Scale bar = 10 μm. **a**, **b** Letters denote statistically significant groupings calculated with Mann–Whitney Rank Sum Test (mean ± s.e.m., *n* = 10; *P* < 0.05). **d** A model for COs/PGN and LCO recognition during symbiosis. The LYR4/CERK1 receptor complex responds to COs and PGN with activation of immunity and symbiosis signalling. In contrast, the LCO receptor activates symbiosis signalling. Activation of symbiosis signalling by LCOs suppresses immunity signalling. Note the co-receptor DMI2 is necessary for COs/PGN and LCO promotion of symbiosis signalling, but is not required for COs/PGN induction of immunity signalling

the recognition of mycorrhizal-produced LCOs[18], a role analogous to its function during rhizobial associations, or alternatively, *NFP* may act in concert with other receptors during recognition of COs.

Arbuscular mycorrhizal fungi appear to be able to promote symbiosis signalling in *M. truncatula* through a combination of COs and LCOs. The additional presence of COs in mycorrhizal fungal exudates allows activation of the symbiosis signalling pathway even when perception of LCOs is restricted and based on the fact that *nfp* mutants show no LCO responses, but maintain wild-type levels of mycorrhizal infection, suggests that CO activation of the symbiosis signalling pathway is sufficient to sustain this symbiosis. Arbuscular mycorrhizal fungi show enhanced production of at least short chain COs in response to the plant signal strigolactone[17], suggesting that COs may be actively produced by these fungi as signals to the plant. *M. truncatula* has many LysM-RLKs in addition to those studied here and at least some of these additional receptors bind LCOs[67]. Hence, multiple receptors may contribute to mycorrhizal recognition of COs, LCOs and possibly other signalling molecules and this may explain the remaining fungal colonisation of the *nfp/Mtcerk1* double mutant. It is possible that a similar situation exists during colonisation by rhizobial bacteria, with both PGN and LCOs able to activate symbiosis signalling.

It has previously been thought that differential chitinaceous molecules show different activities: short chain COs, such as CO4 and CO5 were demonstrated to activate symbiosis signalling[17,20], while long chain COs, such as CO8, induced immunity signalling[2,40]. Such a model implied different perception of COs of different chain lengths and the discrimination of symbiotic and pathogenic fungi through such differential recognition. Here we

show that this separation of CO molecules does not exist: all COs from CO4 to CO8 activate both symbiosis and immunity signalling, at least in *M. truncatula*. CO8 shows the greatest level of elicitation[7,13] and consistently the CO receptors show preferential binding to CO8 over CO4. Promotion of symbiosis signalling[17,20] and immunity signalling[7,8] and the suppression of immunity signalling[36] by CO4 appears contradictory, but it may be that the nature of the response is conditional on the status of the plant at the time of elicitation.

Plant perception of COs and PGN act as a surveillance mechanism for the recognition of fungi and bacteria within the vicinity of the root, but the presence of these elicitors does not differentiate symbiotic from pathogenic microorganisms. Consistently, the plant also does not appear to discriminate between symbiosis versus immunity responses based on CO or PGN perception alone. Our work has focused on *M. truncatula*, but parallel research points at similar situations in other species: silencing of LysM RLKs in *Parasponia* and tomato is detrimental to mycorrhizal colonisation[68,69]; *cerk1* mutants of rice show delayed colonisation of mycorrhizal fungi and reduced responses to CO4[21,22] and *CERK1* of banana has roles in both symbiosis and immunity[25]. LCOs appear to play a pivotal role in symbiosis, with the capability to suppress immunity signalling[36,70] and enhance symbiosis signalling. We suggest that a combination of COs and LCOs promotes a symbiotic outcome, while perception of COs alone preference immunity (Fig. 7d). Critical in this decision-making process is the receptor-kinase DMI2, that promotes symbiosis signalling in response to both COs and LCOs, but appears to play no role in the activation of immunity signalling. The decision to enter a symbiotic interaction may result from the activation of not only symbiosis signalling, but also

other parallel signalling processes and the demonstration that the karrikin receptor is required for mycorrhizal colonisation in rice[71] highlights that additional signalling functions may occur in the plant. Our work reveals that plants choose to encourage or restrict fungal colonisation, not based on the recognition of single signalling molecules, but rather through an integration of the mix of signalling molecules perceived.

## Methods

**Plant materials and growth conditions.** *M. truncatula* cv. Jemalong A17 and R108 were used as wild type, mutants *nfp* (*nfp-1*), *lyk3* (*hcl-1*), *dmi1-2*, *dmi2-3* and *dmi3-1* were reported previously[26,72–74]. The *Tnt1* transposon insertion mutants *lyr4-1* (NF10265), *lyr4-2* (NF15280), *nfp* (NF7796) and *Mtcerk1* (NF16753) were obtained from the Samuel Roberts Noble Foundation collection[7,75]. Yellow Cameleon (YC) 3.6 was incorporated into these mutants by crossing with an existing stable *M. truncatula* YC3.6 line.

*M. truncatula* seeds were treated with concentrated sulfuric acid (Thermo Fisher Scientific) for 10 min, followed with four times wash in sterilized water. The seeds were then surfaced sterilized in 10% sodium hypochlorite for 2 min and rinsed in sterilized water five times. The sterilized seeds were transferred to 1.5% water agar plates and kept in the cold for 2 days and then grown on room temperature overnight before germination. Seedlings were then either grown on medium or cultivated in a 1:1 mixture of sterilized aluminium silicate and sharp sand (Oil-dri Uk Ltd and Berrycroft Horticulture). Plants were incubated in controlled environment rooms at 22 °C and 80% humidity (16 h photoperiod and 300 μE m$^{-2}$ s$^{-1}$). *Arabidopsis* wild type Col-0 and *cerk1-2*[12] were grown in a growth room at 21 °C with a 10 h photoperiod. Five-week old *N. benthamiana* was grown in a glasshouse for protein expression.

**Nuclear calcium imaging.** For calcium analyses *M. truncatula* seedlings were grown on Buffered Nodulation Media (BNM) agar with 100 nM AVG (Sigma-Aldrich) until lateral roots emerged. All calcium analyses were performed on lateral roots since these have been demonstrated to show greater sensitivity to symbiotic signals than the primary root[20]. For the chitinase inhibition assay, *M. truncatula* lateral roots were treated with 0.1% DMSO or 1 mM Acetazolamide (Sigma-Aldrich; dissolved in DMSO) for 30 min before challenging with different concentrations of CO8 to test nuclear calcium oscillations. The lateral roots were fixed in a small chamber made on a cover glass using vacuum grease, with 500 μl of BNM buffer. The roots were then treated with COs, LCOs, flg22 or PGN, at the concentrations indicated. Recordings were collected on an inverted epifluorescence microscope (model TE2000; Nikon). Yellow cameleon YC3.6 was excited with an 458-nm laser and imaged using emission filters 476–486 nm for CFP and 529–540 nm for YFP. The calcium images were collected every 5 s with 1 s exposure and analyzed using Metafluor (Molecular Devices). Calcium traces used the intensity ratio of YFP to CFP. For better visualization of calcium signals, the traces were flattened to a single axis by subtracting the values with moving average as the following formula:

$$S_f = S_o - MA$$

where $S_f$ is the flattened signal, $S_o$ is the original signal, and MA is the moving average of the value. For each element $i$, MA is calculated as:

$$MA = \begin{cases} \sum_{j=1-w}^{i+w} x_j/m & \text{if } i > w \\ \sum_{j=1}^{i+w} x_j/(w+i) & \text{else} \end{cases}$$

where $m$ is the arbitrary number of points to calculate the average, $w = [m/2]$ and $x_j$ is the signal at point $j$.

**Plasmid construction and complementation.** All the constructs used in this research were created using a golden gate cloning strategy[76]. For *lyr4* and *Mtcerk1* complementation, the constructs containing synthesized coding sequences of *LYR4* and *MtCERK1* with 35S terminators under the control of *LYR4* and *MtCERK1* promoters were transformed into *lyr4* and *Mtcerk1* mutants expressing YC3.6 by hairy root transformation. Plant roots with strong DsRed fluorescence were selected to measure calcium oscillations after treatment of 10$^{-8}$ M CO8. For *nfp* (NF7796) complementation, a full-length genomic *NFP* construct containing promoter and coding sequences was synthesized and cloned into a destination vector along with DsRed for plant selection. The *nfp* complementation plants carrying transgenic roots, were transferred to aluminium silicate/sand and grown for 3 days prior to inoculation with 3 mL *S. meliloti* strain 1021 at an optical density of OD$_{600}$ = 0.05. Four weeks post inoculation plant roots were washed and nodules were quantified and imaged using a Leica M205FA stereo microscope.

To generate constructs for protein expression in *N. benthamiana*, the synthesized coding sequences of *LYR4*, *MtCERK1*, *LYK3* and *DMI2* were fused with 3xHA or 3xFLAG tag to generate LYR4-HA, *Mt*CERK1-HA, LYK3-HA, DMI2-HA and *Mt*CERK1-FLAG. All constructs were transformed into *Agrobacterium* GV3101 for transient expression in tobacco leaves.

**Mycorrhizal inoculation.** *M. truncatula* wild type and mutant plants were grown in pots (4 × 4 × 4.5 cm$^3$) containing aluminium silicate /sand and inoculated with 200 spores of *R. irregularis* produced by Premier Tech (Québec, Canada). Mycorrhizal colonized roots at the different time points indicated were collected and stained with ink. The grid line intersect method[77] was used to quantify mycorrhizal colonization, roots were cut into small segments and spread randomly in plastic petri dishes in which a grid with 1 cm × 1 cm squares was affixed to the base. One hundred and twenty intersections for each root sample were counted to measure roots with or without mycorrhizal infection on a Leica DM6000 light microscope.

**WGA staining.** *M. truncatula* R108, *Mtcerk1* and *Mtcerk1/nfp* plants were cultivated in aluminium silicate/sand with mycorrhizal spores for 3 weeks. Roots were then immersed in 0.1 M HCL for 2 h and washed with 1× PBS buffer, then stained in WGA-Alexa Fluor 488 dye (Molecular Probes) for 6 h. Images were taken using a Leica DM6000 light microscope or for higher resolution images a Zeiss LSM 780 confocal microscope.

**Gene expression analysis.** *M. truncatula* wild type and mutants were grown on BNM plates with 100 nM AVG for four days, the plants were then carefully transferred to liquid BNM with control DMSO or elicitors (10$^{-7}$ M flg22, 10$^{-7}$ M CO4, 10$^{-7}$ M CO8 or 10$^{-8}$ M *Sm*LCO) for 6 h. We used eight plants for each treatment and three biological replicates for each time point. Plants roots were harvested and frozen in liquid nitrogen. Total RNA was extracted from root tissues using the Qiagen RNeasy Plant mini kit, DNA was removed by treatment of RNA with RNase-free DNase according to the manufacturer's instructions. An aliquot of each total RNA sample was used to determine RNA concentration and purity assessed on the NanoDrop ND-1000 spectral photometer (Peqlab, Erlangen, Germany). RNA integrity was analyzed on the 2100 Bioanalyzer using RNA 6000 Nano LabChip Kits (Agilent Technologies). Library preparation was performed as described for the TruSeq® Stranded mRNA HT kit and sequencing of the library was run on a NextSeq500 sequencing system (Illumina). The resulting reads were quality controlled and mapped against the *M. truncatula* A17 reference genome (Mtv4.0) using Trimmomatic and the STAR RNA-seq aligner and analyzed for differential gene expression in different comparisons using the edgeR statistical package. Fold changes ≥ 2 and False Discovery Rate (FDR) corrected *p*-values ≤ 0.05 were used as thresholds for the genes to be identified as being differentially expressed (Supplementary Data 1). The row values representing the gene expression fold change values for the respective treatment compared to control were hierarchically clustered with the one-minus Pearson correlation option using the average value for each row with the default settings on the GENE-E program, the data for mycorrhizal induced gene expression were retrieved from published data[78]. The raw data from the RNA sequencing has been deposited at the NCBI Sequence Read Archive (SRA) with accession number SRP097705.

For qRT-PCR analysis, *M. truncatula* was grown under the same conditions as that used for production of the RNA-seq material. Plants were treated with COs, PGN, flg22 and *Sm*LCO, at a final concentration of 10$^{-7}$ M COs, 10$^{-7}$ M flg22, 0.4 mg/ml PGN and 10$^{-8}$ M *Sm*LCO. After incubation for 6 h RNA was isolated using the methods described above. One microgram of total RNA was used for cDNA synthesis with an iScript™ cDNA Synthesis Kit (Bio-Rad). Gene expression was determined by an ABI 7500 using a SYBR green PCR master mix (Bio-Rad). The *Vapyrin* (*Medtr6g027840*), *HA1* (*Medtr8g006790*), *MtLYK10* (*Medtr5g033490*) and *D27* (*Medtr1g471050*) were used as symbiosis marker genes while pathogenesis-related proteins *PR10* (*Medtr4g120940*), *PR4* (*Medtr1g080800*), a plant chitinase (*Medtr2g099470*) and a plant glutathione S-transferase *GST* (*Medtr3g467420*) were considered as defense marker genes (all were induced by flg22). Expression data were analyzed from the average of threshold cycle (CT) value, using an *M. truncatula* endogenous *histone* 2 A gene as reference and fold induction calculated for treatment of elicitors relative to treatment with DMSO. The qRT-PCR primers used can be found in Supplementary Table 4.

**Reactive oxygen species.** Leaf discs (4 mm diameter) from 4-week-old *Arabidopsis* Col-0 and *cerk1-2* plants were collected to measure ROS. *M. truncatula* primary roots growing on 1.5% water agar for 5 days were cut into 0.5 cm strips and incubated in 200 μL water in a 96-well plate (Greiner Bio-one) overnight. For the *Sm*LCO inhibition assay, the infection zone area of wild type and mutant primary roots were used and root segments were pretreated with either water or 10$^{-8}$ M *Sm*LCO over different time points before being challenged with 0.4 mg/ml PGN or 10$^{-6}$ M CO8 and/or 10$^{-8}$ M *Sm*LCO. For cycloheximide treatments, *M. truncatula* roots were pre-incubated with 100 μM cycloheximide (Sigma-Aldrich) for 4 h before *Sm*LCO treatment. For chitinase inhibitor treatments, the root segments were pretreated with 0.1% DMSO or 1 mM Acetazolamide for 30 min before addition of elicitors. After incubation, the water was removed from each well and exchanged with 200 μL reaction buffer containing 10$^{-6}$ M CO8 or CO8 with 1 mM Acetazolamide, 10 μg/ml horseradish peroxidase (Sigma-Aldrich) and 20 μM luminol (Sigma-Aldrich) or 0.5mM L-012 (Wako Chemicals, USA) according to experiments performed. The fungal germinated spores exudates (GSE, 10 times concentrated) were used to detect ROS production in *Arabidopsis* leaves and

*M. truncatula* roots. Luminescence was recorded with a Varioskan™ Flash Multi-mode Reader (Thermo Fisher Scientific) at indicated time points.

**MAPK activity assay**. One-week old *M. truncatula* roots grown on modified Fahraeus medium were excised into small segments and incubated overnight in water. The root samples were then treated with $10^{-6}$ M CO8 or DMSO for 10 min, then frozen in liquid nitrogen immediately for protein extraction. The *Sm*LCO inhibition assay was performed at the same treatment as detection of ROS production, roots were incubated with water or *Sm*LCO for 30 min, and then treated with $10^{-6}$ M CO8 and/or $10^{-8}$ M *Sm*LCO for 10 min. The root protein was homogenized in an extraction buffer containing 50 mM HEPES-KOH (pH 7.5), 150 mM KCl, 1 mM EDTA, 0.5% Triton-X100, 1 mM DTT, complete protease inhibitors (Roche) and phosphatase inhibitors (Roche). Equal amounts of total protein were electrophoresed on 10% SDS–PAGE. An anti-pERK antibody (Cell Signalling Technology, Cat#4370, 1:3000 dilution) was used to determine phosphorylation of MPK3 and MPK6 by western blot, a duplicate blot was used to detect *M. truncatula* tubulin to show the total protein levels using anti-tubulin (Sigma-Aldrich, Cat# T5168, 1:5000 dilution).

**Chitin binding assay**. *Agrobacterium* GV3101 transformants carrying LYR4-HA, *Mt*CERK1-HA and DMI2-HA constructs were suspended in the infiltration buffer containing 10 mM MgCl$_2$, 10 mM MES-KOH (pH 5.6) and 100 μM acetosyringone (Sigma-Aldrich), then mixed 1:1 with GV3101 containing a P19 construct. The mixture of strains were infiltrated into *N. benthamiana* leaves and left for 12 h. Total protein was then extracted with cold lysis buffer (50 mM HEPES-KOH pH 7.5, 150 mM KCl, 5% Glycerol, 1 mM EDTA, 0.5% Triton-X 100, 2 mM DTT, complete protease inhibitors and 2% PVPP).

Biotinylated conjugates of CO4 and CO8 were prepared as described previously[51]. In brief, CO8 and CO4 reacted with the oligo (ethylene glycol) linker (OEG-linker) to form an oxime linkage at the reducing end. The resulting conjugate containing a free thiol was used in a reaction with a biotin moiety. The biotinylated CO4 and CO8 were then immobilized to streptavidin-coated beads and stored at 25 μM concentration in 50% acetonitrile/water at 4 °C. Conjugates were immobilized on Dynabeads® MyOne™ Streptavidin C1 beads (LifeTechnologies) as follows. Beads were washed three times with PBS, pH 7.2, 0.01% Tween-20 buffer by using a DynaMag™-2 magnetic rack. Binding capacities of Dynabeads were estimated according to the manufacturer's instructions. The biotinylated ligand was diluted in PBS, pH 7.2, 0.01% Tween-20 buffer and added to the beads in approx. two-fold molar excess of the estimated binding capacity. After 30 min incubation at room temperature while shaking, beads were washed three times with PBS pH 7.2, 0.01% Tween-20 and subsequently reconstituted in PBS pH 7.2 0.01% Tween-20 to a final concentration of 10 mg/ml. For the binding assay, 20 μL of CO4 or CO8 beads were washed 3 times by lysis buffer and then incubated with the extracted proteins for 3 h. The Dynabeads® MyOne™ Streptavidin C1 beads without crosslinked COs were used as a control. For the PGN binding assay, 50 μL PGN (0.8 mg/ml) was centrifuged to isolate a soluble fraction in the supernatant and an insoluble pellet fraction. The PGN pellet was then mixed with extracted proteins and allowed to bind for 3 h. After washing six times with wash buffer (50 mM HEPES-KOH pH 7.5, 400 mM KCl, 1 mM EDTA, 0.5% Triton-X 100, 1 mM DTT, complete protease inhibitors), the bound protein was recovered by boiling either beads or PGN pellet with SDS loading buffer. For elicitor competition assays, equal levels of proteins were incubated with CO4 and CO8 beads or PGN pellet and following the wash steps, the bound proteins was eluted by adding 100 μM *Sm*LCO, 100 μM COs or PGN soluble fraction. The anti-HA antibody (Sigma-Aldrich, Cat#12013819001, 1:3000 dilution) was used to detect the protein bound to COs beads and PGN pellet.

**Cell death assay in tobacco leaves**. *Agrobacterium* GV3101 transformants carrying the respective construct were diluted to OD600 = 0.6 to infiltrate into fully expanded leaves of *N.benthamiana* as described above. The plants were then covered in a transparent dome to maintain high humidity and cell death induction was recorded between 24 and 72 h. After 12 h post-infiltration, leaf punches from each infiltrated area were collected to extract total protein. The expression level of each protein was detected by western blotting with anti-HA and anti-FLAG antibodies (Sigma-Aldrich, A8592-2MG, 1:5000 dilution). To assess cell death leaves were stained with Trypan blue that marks dead cells.

**Pathogen infection**. *Phytophthora palmivora* was grown on V8 juice agar medium for 7 days until mycelium were fully expanded over the whole plate. The plates were then kept in a fume hood for 24 h, to dry the medium. 10 ml sterilized cool water was poured on each plate and kept for 1 h to release zoospores. The concentration of spores was quantified on a hemacytometer. *M. truncatula* wild type and mutant seedlings were grown on 0.8% agarose plates with or without $10^{-7}$ M *Sm*LCO for 1–2 days and root tip regions were inoculated with either $1 \times 10^5$/ml *P. palmivora* spores or a mixture of $10^{-7}$ M *Sm*LCO and spores at $1 \times 10^5$/ml concentration. To quantify *P. palmivora* growth, 30 seedlings for each ecotype 48 h post-inoculation were used to measure the lesion size and this was normalized to the individual root length. Uncolonised roots were not included in the analysis. The same root samples were then frozen in liquid nitrogen to extract RNA for qRT-PCR analysis using *P. palmivora* EF1a gene relative to the *Medicago* histone *H2A* housekeeping gene.

**MALDI-TOF mass spectrometry**. To detect the purity of CO4 and CO8, 1 μL droplets of CO4 or CO8 were mixed with 1 μL 2,5-dihydroxybenzoic acid (10 mg/ml, Sigma-Aldrich) in TA30 solvent (30:70 [v/v] acetonitrile: 0.1% TFA in water, Sigma-Aldrich). Aliquots of 1 μl of this sample were then spotted onto a MTP AnchorChip 384 MALDI target plate (Bruker, UK) and analyzed for the intensity of COs corresponding to the exact molecular weight by MALDI-TOF mass spectrometry with the Autoflex speed system (Bruker, UK). Data processing was conducted using FlexAnalysis (Bruker, UK). For chitinase inhibitor treatments, *M. truncatula* lateral roots were pretreated with either 0.1% DMSO or 1 mM Acetazolamide for 30 min before incubations with CO8. One microlitre of sample from the incubation solution was taken to mix with 2,5-dihydroxybenzoic acid matrix to detect the intensity of CO8 at indicated time points.

**Reporting summary**. Further information on research design is available in the Nature Research Reporting Summary linked to this article.

## Data availability

The RNA-seq data have been deposited in GenBank with accession number SRP097705 [https://www.ncbi.nlm.nih.gov/bioproject/PRJNA363004]. Data and materials supporting the observations in this study are available from the authors upon request. The source data underlying Figs. 1a, c, 2a, b, 3a–d, 4b, e, 5a–c, 6a–c, 7a, b and Supplementary Figs. 1a, b, 2a–c, 3a, b, 4a, b, 6b–e, 8b, c, 9a–c, 10a–c, 11a–c, 12a, b are provided as a Source Data file.

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

## Acknowledgements

We thank Jiangqi Wen and Kirankumar S. Mysore for providing *M. truncatula* mutants from their core collection, Uta Paszkowski for providing *R. irregularis* germinated spore exudates and Sebastian Schornack for providing *Phytophthora palmivora* spores. This work was supported by the Bill and Melinda Gates Foundation as Engineering the Nitrogen symbiosis for Africa, OPP1028264, the Biotechnology and Biological Sciences Research Council as BB/J004553/1 and BB/K003712/1, and the Danish National Research Foundation as DNRF79. SF acknowledges S. Pradeau for her technical assistance, the mass spectrometry facilities at ICMG (FR 2607), the chromatography facilities at CERMAV as well as Carnot Institut PolyNat and Labex Arcane (ANR-11-LABX-0003-01).

## Author contributions

F.F., J.H.S. and G.E.D.O. conceived and designed the experiments. F.F. and J.H.S. performed the research. T.L. and G.V.R. analyzed RNA-seq data. S.F. synthesized CO and LCO molecules. K.G., M.B.T. and K.R.A. prepared CO4 and CO8 beads. A.G. supported the work on *P. palmivora*. F.F. and G.E.D.O. wrote the paper with comments from Z.B., J.S. and S.R.

## Competing interests

The authors declare no competing interests.
