## [Peer Review File · Nature Communications]

Reviewers' comments:

Reviewer #1 (Remarks to the Author):

This paper investigates perception of the dual PAMP/symbiosis chitoooligosaccharide signals of fungi. The authors find that CO8, a well studied PAMP, is necessary for mycorrhizal association and also for the characteristic symbiosis-related nuclear calcium spiking effect. Both responses required the known chitin receptor MtCERK1, with lesser contributions from the LysM receptors NFP and LYR4, and both MtCERK1 and MtLYR4 bound COs in vitro. COs induced symbiosis genes dependent on the three LysM receptors. The paper deals briefly with the known suppression of defences responses by LCOs, but this is not provide significant progression over what is in the literature. Overall, while it is a highly competent and well written study, I find it rather incremental in its advances.

Intro para 2 "Recognition of CO8...in Arabidopsis...involves a heteromer of two LysM-containing receptors". There is an alternative model from JieJie Chai which has a homodimer of CERK1 as necessary and sufficient for chitin perception DOI: 10.1126/science.1218867. I am not sure this model has been falsified, and hence should be kept in mind during interpretation of the current results.

Intro para 3 "One lacking kinase activity NFP in *M. truncatula*..." maybe rewrite this, it's unclear. Please use line numbers in future, makes it far easier for the reviewer.

The introduction would benefit from some description/explanation of trichoblasts and atrichoblasts for the general reader.

Fig. 1a What approximate concentration is 0.8 mg/ml PGN? At what concentration is spiking lost?

Fig. 1 is convincing and shows that CO8 (which is the PAMP chitin) induces calcium spiking in trichoblasts that are associated with symbiosis signaling. Surprisingly, PGN does also.

I have reservations about the use of ROS and MAPK as outputs of immune signaling - the authors will know that these are induced by many receptors and are not specific outputs, please comment?

Fig 2 is solid but could maybe go to supplementary. I agree that AM fungi are likely to produce CO8 (the alternative seems far more unlikely).

DMI2 was required for CO8 calcium spiking...this seems like potentially a very important result to me, and I think this aspect has not been investigated strongly enough in the current MS.

"To assess if this CO8 receptor complex"...you are not detecting a complex, change to "members of the complex".

"We conclude that a single receptor complex, most likely containing"...delete this sentence, it runs far in advance of the data that you have collected. The data contain only genetic information that are limited by the mutants used.

Fig 3. In the context of the co-submitted paper, the fact that NFP is not required for CO4/8 Ca²⁺ spiking is surprising, at least to me. However, in Fig 4b we see that NFP is required for gene changes induced by CO4/8 - please could you comment on this?

Fig 4 is very nice and clear, although limited of course by the fact that this is an in vitro assay.

Fig 5 One should bear in mind with this type of experiment that we are looking at the gene expression of both trichoblasts and atrichoblasts (in what proportions?).

Fig 5a what does this figure show/how is it to be interpreted? Actually the text is quite descriptive, but the graphic itself is difficult to read (I know it's a standard representation, just not a very useful one). Very surprised at the overlap of all treatments with flg22 (~110 genes shared with SmLCO).

Fig 5b I regret that the flg22 genes were removed, because I think it is quite possible that the LCO receptors suppress immunity, and this information is now lost. In this respect, I would also liked to have seen a dmi2 mutant.

Fig 5b the figure is accurately described in the text, but the gene identities are not, perhaps a lost opportunity. That makes 5 c and 5 d very limited. However, I agree with the conclusions about the roles for the receptor genes in transcriptional responses. Is there an explanation of the cladogram on the LHS of the figure, I didn't find one.

"The combination of RNAseq and qRT-PCR reveals significant overlap between treatments"...I am not very comfortable with this statement. Firstly, the flg22 signal was removed to force the data closer together, and secondly there is no point of reference (some other root-perceived signal). Please respond or modify the statement.

Fig 6, again, I remind the authors that ROS and MAPK assays are not immunity per se, but immunity-associated responses that are not specific to immunity. Is it possible to devise a pathogen assay with a root pathogen so that immunity itself can be measured? Also, I would like to see a dmi2 mutant used in these assays. These data do not significantly extend what is known since Liang et al. 2013 (Stacey).

I also suggest that the leaf assay in which co-expression of LjNRF5 and LjNFR1 causes cell death in *N. benthamiana* might be another avenue to explore these phenomena.

The PGN data are oddly described in this MS, tacked on into the final paragraph of the results...would prefer it to be integrated

Discussion - I found this rather long-winded and for the most part reiterates the results. The most important insight from this study is that CO perception acts redundantly in identification of fungal pathogens and symbionts, but how the plant interprets these signals accurately is very unclear.

Check (a)trichoblasts spelling.

Reviewer #2 (Remarks to the Author):

The manuscript by Feng and colleagues presents the results of an extensive investigation, combining a number of approaches to study the responses of the model legume *M. truncatula* to different chitin-related molecules that play a role in the perception of arbuscular mycorrhizal fungi (beneficial mutualistic biotrophs). The main conclusions focus on the role of LCOs as veritable symbiotic signals, compared to COs of different length, which are all envisaged as defense elicitors. The latter point represents the most innovative aspect, in open contrast with the current view of longer chain COs (CO8) as defense elicitors, and short chain COs (CO4-CO5) as symbiotic signals.

I only have a few specific critics to the result presentation, including a couple of obvious contradictions, which I list below, but I first have to raise a few concerns about the rationale and – consequently – result interpretation and conclusions.

The manuscript rationale is based on a number of assumptions, which emerge throughout the text.

I unfortunately have to disagree with a few of them. In particular:

1) That nuclear calcium spiking is an univocal hallmark of symbiotic signal transduction. There is evidence that mutants in the nuclear calcium-dependent kinase DMI3 and other members of the so-called common symbiotic signaling pathway are affected in their responses to non-symbiotic microbes, such as parasitic nematodes and parasitic plants, or necrotrophic and hemibiotrophic fungal pathogens. On this basis one cannot exclude that nuclear calcium signals are involved in such interactions too, and this should be verified carefully.

2) That the application of purified molecules can mimic the plant-directed signals generated by each symbiont. This aspect is particularly critical, because the authors are aware of the synergistic and complex chemical message that a mix of several different molecules may generate. Consequently, the dissection of plant responses based on single molecule application is a risky strategy that can be reasonable to support direct conclusions (e.g. CO8 triggers nuclear calcium spiking) but may have limited biological meaning if the conclusions extend too far from the experimental result (e.g. CO8 have or not a role in symbiosis). What evidence do we have that the application of one purified molecule to root cells that are growing under partial nutrient starvation is activating one ligand-specific pathway and not triggering a crosstalk with other pathways that start from structurally very similar LysM receptors? In fact, the matter of receptor complex composition is one huge open question in this field and trying to draw major conclusions in the absence of this piece of knowledge is, in my opinion, largely speculative.

3) That AM fungi produce CO8. I find particularly puzzling the importance given to CO8, a class of chitin oligomers that has not been characterized so far in AM fungal exudates and whose presence is depicted by the authors as a 'very likely [...] byproduct of chitinaceous fungal cell walls' (page 5). In fact, it is reasonable to speculate that chitin-related fungal signaling molecules derive from direct biosynthetic pathways rather than the dismantling of the crystalline chitin fibers in the wall (which is quite unlikely, in my opinion). Whatever their metabolic or catabolic origin, it remains to be demonstrated that this class of COs is produced by AM fungi during their interaction with the host plant.

Specific comments:

Page 3. The second paragraph states that "CO4 can activate immunity signaling". The last paragraph in the same page reads "the plant actively suppresses immunity and this occurs following recognition of both CO4 and LCOs". Is this not contradictory?

The role of LCOs as shifters of the plant response toward symbiosis appears difficult to accept in the light of the normal AM phenotype of *nfp* mutants. The proposal that other LysM receptors may replace the *nfp* function remains to be demonstrated and falls within the abovementioned lack of information on the composition of receptor complexes.

On the same line, the conclusion that LCOs 'play an essential role in directing a symbiotic outcome' (page 11) is in contrast with the observations in rice, where LCOs do not appear to be perceived by the root. My feeling is that the use of legumes is of limited help for the search of AM-specific signaling processes, due to the ability of these plants to interact with both fungal and bacterial symbionts. Overlaps and crosswalks between the respective signaling pathways cannot be excluded (and are also justified by their evolutionary relationship), which becomes especially critical when the roots are challenged with purified molecules (see my previous comment).

The tests of calcium spiking activation in trichoblasts, presented in figure 1, is of limited interest in the frame of AM interactions, where this cell type is not involved in colonization. Other figures also consider atrichoblasts. I wonder why this important initial test was not done on atrichoblasts too. My personal interpretation of the graphs in fig 1 is that trichoblasts are more responsive to nod factor-like molecules than COs, which is perfectly in line with the depicted role of trichoblasts and atrichoblasts as targets of rhizobia and AM fungi, respectively.

The comparison of gene expression patterns reveal a limited overlap between gene expression patterns induced by CO8 and CO4, while CO4 do activate more symbiosis- than defense-related genes compared to CO8. Taking into account my critic on the use of single molecules - or rather on the interpretation and biological significance of the results obtained from single molecule treatments – and our lack of knowledge about the composition of the receptor complex(es) that regulate such responses, I find it difficult to agree with the author conclusions.

We are dealing with a phosphate starved plant that is exposed to structurally very similar chitin-based molecules, which can be bound by several receptors with high structural similarity and variable affinities for each of these ligands. I would assume that under such conditions, whatever chitin-backbone molecule may at least partially activate non-specific pathways alongside its own, leading to vast overlaps in the resulting gene expression patterns. For this reason I would put the focus on differences (which may reveal specificities) rather than similarities between the expression patterns.

The last paragraph of page 10 strongly points at the role of LCOs as true symbiotic signals, but what about rice, where LCOs are not active? As I mentioned above, I have the impression that studying these responses in legumes is not ideal, at least until we have clarified in depth what receptors rule AM interactions in plants (like rice) where there is no possible confusion with nod factor signaling.

The use of arabidopsis leaf treatments with *R. irregularis* exudate appears to me like a very unnatural condition. I can hardly understand what conclusions can be drawn from such an experiment, where a non host organ from a non host plant is exposed to the molecules released by a root-interacting symbiotic fungus.

In the same paragraph, DMI2 is presented as a critical member of the CSSP for the induction of symbiosis-specific responses. Nevertheless, Fig. 3 shows that the spiking response induced by CO8 and PGN is absent in *dmi2* mutants. Does this not indicate a role for DMI2 in defense? And consequently, does this not suggest that nuclear calcium spiking may also be present in the response to pathogens (also see my previous remark)?

In fact this concept contrasts with figure 7d, where DMI2 is represented to have roles in both symbiotic and defense signaling

Page 10, second paragraph, states that “What does appear to differentiate symbionts from pathogens is the presence of LCOs”. Has LCO production been demonstrated for all phylogenetic groups of AM fungi? And in what quantity compared to COs and compared to the concentrations used in the experiments? Again, how about non-legumes like rice, where LCOs are inactive?

Figure 2. Why is panel d in this figure? It does not appear to have any relation with the figure title.

Figure 4. Panel c should have a PGN label for uniformity with the remaining panels

Figure 7. Panel d: As mentioned above, the role of DMI2 does not correspond to what is stated in the text (and in suppl. Figure 2). Furthermore, where does CO4 position in this model?

Suppl. Fig. 1. Panel c does not appear to belong to this figure (no connection with the figure title)

Suppl. Fig. 2.” DMI2 is not required for CO8-induced defense responses”. Then why is DMI2 in the scheme of figure 7d?

Suppl. Fig. 3. Why did the authors choose to show the spiking patterns of the few PGN treated cells in panel e, when they did not for other treatments?

Suppl. Fig 5. I see important differences in the pattern of defense-related gene regulation between WTCO4 and WTCO8 (see my comment above).

Reviewers' comments:

Reviewer #1 (Remarks to the Author):

This paper investigates perception of the dual PAMP/symbiosis chito oligosaccharide signals of fungi. The authors find that CO8, a well studied PAMP, is necessary for mycorrhizal association and also for the characteristic symbiosis-related nuclear calcium spiking effect. Both responses required the known chitin receptor MtCERK1, with lesser contributions from the LysM receptors NFP and LYR4, and both MtCERK1 and MtLYR4 bound COs in vitro. COs induced symbiosis genes dependent on the three LysM receptors. The paper deals briefly with the known suppression of defences responses by LCOs, but this is not provide significant progression over what is in the literature.

Response: We have now added considerably more data on LCO suppression of immunity, showing the genetic dependencies of this response (interestingly only *NFP* and not the Sym pathway are important), showing that LCOs promote pathogen colonisation and that this effect is dependent on novel protein synthesis. We believe that this new data creates significant novelty in this aspect of the work, which has not been observed previously.

Intro para 2 “Recognition of CO8...in Arabidopsis...involves a heteromer of two LysM-containing receptors”. There is an alternative model from JieJie Chai which has a homodimer of CERK1 as necessary and sufficient for chitin perception DOI: 10.1126/science.1218867. I am not sure this model has been falsified, and hence should be kept in mind during interpretation of the current results.

Response: Our data implies two receptors functioning in CO8 recognition, which agrees with a two receptor model in *Arabidopsis* (Cao *et al.*, 2014) and the two receptors in rice, OsCEBiP and OsCERK1. This is also consistent with two receptors playing a role in LCO perception. Hence, while we are not attempting to refute the model of Chai, we believe that the heterodimer model proposed by others is a much better fit for what we have found in *Medicago*.

Intro para 3 “One lacking kinase activity NFP in *M. truncatula*...” maybe rewrite this, it’s unclear. Please use line numbers in future, makes it far easier for the reviewer.

Response: We have now removed this terminology.

The introduction would benefit from some description/explanation of trichoblasts and atrichoblasts for the general reader.

Response: We followed the reviewer’s suggestion and added the explanation in the introduction.

Fig. 1a What approximate concentration is 0.8 mg/ml PGN? At what concentration is spiking lost?

Response: PGN is a suspension, only some of the PGN powder dissolves in water and it is impossible to measure the exact concentration, hence we state (as do other authors: Willmann *et al.*, 2011; Liu *et al.*, 2012) the amount added relative to volume (0.8 mg/ml). We found 0.4 mg/ml activates a very weak response (only one cell responding in 16 cells tested).

Fig. 1 is convincing and shows that CO8 (which is the PAMP chitin) induces calcium spiking in trichoblasts that are associated with symbiosis signaling. Surprisingly, PGN does also.

Response: We believe that this demonstrates that sugar-based MAMPs may have general functions in promotion of both immunity and symbiosis.

I have reservations about the use of ROS and MAPK as outputs of immune signaling - the authors will know that these are induced by many receptors and are not specific outputs, please comment?

Response: ROS and MAPKs are intrinsic components of immunity signalling and have been used for many years, by many researchers, as reporters for immunity signalling (Day *et al.*, 2001; Miya *et al.*, 2007; Cao *et al.*, 2014). We have complemented our work with induction of markers genes for defences and now have also added data on a root pathogen. Hence, we are not relying on these outputs alone to build our case. The ROS and MAPK assays agree with defence reporters and with the pathogen work. Hence, we believe that these are valuable assays for assessing immunity signalling and we are certainly not alone in using these assays, we follow many researchers of immunity that use these assays as standard.

Fig 2 is solid but could maybe go to supplementary. I agree that AM fungi are likely to produce CO8 (the alternative seems far more unlikely).

Response: We have moved this figure to supplementary.

DMI2 was required for CO8 calcium spiking...this seems like potentially a very important result to me, and I think this aspect has not been investigated strongly enough in the current MS.

Response: We have added new data on DMI2 (and other components of the Sym pathway) in Fig.6, Supplementary Figures. 6 and 9. We have also emphasised the point in the discussion that the presence of DMI2 could be the discriminator for the activation of symbiosis versus immunity.

“To assess if this CO8 receptor complex”...you are not detecting a complex, change to “members of the complex”.

Response: We have changed the text accordingly.

“We conclude that a single receptor complex, most likely containing”...delete this sentence, it runs far in advance of the data that you have collected. The data contain only genetic information that are limited by the mutants used.

Response: We have changed the text accordingly.

Fig 3. In the context of the co-submitted paper, the fact that NFP is not required for CO4/8 Ca²⁺ spiking is surprising, at least to me. However, in Fig 4b we see that NFP is required for gene changes induced by CO4/8 - please could you comment on this?

Response: We believe that NFP acts primarily as an LCO receptor and therefore we do not find it surprising that NFP is not essential for CO responses. To us the partial dependency of the CO8 transcriptional response on *NFP* is the surprising result. However, the LCO

transcriptional response is almost completely dependent on *NFP*, while the CO8 response is only partially dependent. We now provide new data showing evidence for signalling downstream of *NFP*, that is independent of the Sym pathway. Hence, we would expect to see effects of *NFP* on the RNA-seq that are independent of the Sym pathway and this likely explains these differences.

Fig 4 is very nice and clear, although limited of course by the fact that this is an in vitro assay.

Fig 5 One should bear in mind with this type of experiment that we are looking at the gene expression of both trichoblasts and atrichoblasts (in what proportions?).

Response: This is a very valid point and we highlight this fact in our description of the effect of *lyr4* on this transcriptomic response.

Fig 5a what does this figure show/how is it to be interpreted? Actually the text is quite descriptive, but the graphic itself is difficult to read (I know it's a standard representation, just not a very useful one). Very surprised at the overlap of all treatments with *flg22* (~110 genes shared with *SmLCO*).

Response: We believe it is useful to show the relative overlaps between these different treatments and we are not aware of a better way of showing this. However, we have moved this data to supplementary.

Fig 5b I regret that the *flg22* genes were removed, because I think it is quite possible that the LCO receptors suppress immunity, and this information is now lost. In this respect, I would also like to have seen a *dmi2* mutant.

Response: We have now redrawn the heatmaps to include all genes induced by *flg22*, CO4, CO8 and *SmLCO*. *Nfp* shows an altered response to *SmLCO*: loss of most of the genes induced in wild type, but gain of some gene induction, not normally activated in wild type. Some of these gains are also induced by *flg22*, but we do not see a recapitulation of the genes induced by both *flg22* and CO8. Hence, there is some evidence that *NFP* is involved in suppressing gene induction and our new data demonstrates that *NFP* does suppress immunity in response to LCO treatment.

Fig 5b the figure is accurately described in the text, but the gene identities are not, perhaps a lost opportunity. That makes 5 c and 5 d very limited. However, I agree with the conclusions about the roles for the receptor genes in transcriptional responses. Is there an explanation of the cladogram on the LHS of the figure, I didn't find one.

Response: We have included the gene annotations in Supplementary Table 4 that show all the differentially expressed genes responding to the different elicitors. It is not possible to include the gene identifiers in Fig.4a and at the same time show the scale of this transcriptome response. We have to choose between showing the full response versus showing a smaller set of genes in which we could include the identifiers. We have tried to find this balance by showing the full set of the response in Fig.4a, a subset of the response (including gene identifiers) in Supplementary Figure 7 and examples of individual genes using qRT-PCR in Fig.4 and Supplementary Figure 6.

“The combination of RNAseq and qRT-PCR reveals significant overlap between treatments”...I am not very comfortable with this statement. Firstly, the flg22 signal was removed to force the data closer together, and secondly there is no point of reference (some other root-perceived signal). Please respond or modify the statement.

Response: We have modified the text to state: Our work reveals that CO8 activates genes associated with both immunity and symbiosis and this correlates with its ability to activate both immunity and symbiosis signalling.

Fig 6, again, I remind the authors that ROS and MAPK assays are not immunity per se, but immunity-associated responses that are not specific to immunity. Is it possible to devise a pathogen assay with a root pathogen so that immunity itself can be measured? Also, I would like to see a *dmi2* mutant used in these assays. These data do not significantly extend what is known since Liang et al. 2013 (Stacey).

Response: We have now significantly expanded this aspect of the work, showing a mutant analysis of the LCO suppression of symbiosis signalling and including the analysis of a root pathogen, *Phytophthora palmivora*. *DMI2* is not required for LCO suppression or immunity and as previously described has no defect in colonisation by this pathogen.

I also suggest that the leaf assay in which co-expression of LjNRF5 and LjNFR1 causes cell death in *N. benthamiana* might be another avenue to explore these phenomena.

Response: We have included this experiment in Supplementary Figure 9. The conclusion is that overexpression of NFP has no effect on MtCERK1-induced cell death in *N.benthamiana*.

The PGN data are oddly described in this MS, tacked on into the final paragraph of the results...would prefer it to be integrated

Response: The manuscript is primarily structured around an analysis of arbuscular mycorrhizal associations. We believe the PGN data is interesting, in that it shows that PGN, like CO8, acts as a dual symbiosis and immunity elicitor. However, unlike mycorrhization, we do not see an impact of the CERK1 or LYR4 receptors on rhizobial colonisation. Hence, at this stage it is not possible to explain the relevance of the PGN data for bacterial colonisation. In order to maintain coherence in the manuscript we chose to describe the PGN data separately to CO8 and we would prefer to maintain this structure as we believe it is easier to follow for readers.

Discussion - I found this rather long-winded and for the most part reiterates the results. The most important insight from this study is that CO perception acts redundantly in identification of fungal pathogens and symbionts, but how the plant interprets these signals accurately is very unclear.

Response: We have shortened and focused the discussion.

Check (a)trichoblasts spelling.

Response: Corrected

Reviewer #2 (Remarks to the Author):

The manuscript by Feng and colleagues presents the results of an extensive investigation, combining a number of approaches to study the responses of the model legume *M. truncatula* to different chitin-related molecules that play a role in the perception of arbuscular mycorrhizal fungi (beneficial mutualistic biotrophs). The main conclusions focus on the role of LCOs as veritable symbiotic signals, compared to COs of different length, which are all envisaged as defense elicitors. The latter point represents the most innovative aspect, in open contrast with the current view of longer chain COs (CO8) as defense elicitors, and short chain COs (CO4-CO5) as symbiotic signals.

I only have a few specific critics to the result presentation, including a couple of obvious contradictions, which I list below, but I first have to raise a few concerns about the rationale and – consequently – result interpretation and conclusions.

The manuscript rationale is based on a number of assumptions, which emerge throughout the text. I unfortunately have to disagree with a few of them. In particular:

1) That nuclear calcium spiking is an univocal hallmark of symbiotic signal transduction. There is evidence that mutants in the nuclear calcium-dependent kinase DMI3 and other members of the so-called common symbiotic signaling pathway are affected in their responses to non-symbiotic microbes, such as parasitic nematodes and parasitic plants, or necrotrophic and hemibiotrophic fungal pathogens. On this basis one cannot exclude that nuclear calcium signals are involved in such interactions too, and this should be verified carefully.

Response: We value these thoughts from the reviewers, but we respectfully disagree with the reviewer. Many studies in many species of plants, legumes and non-legumes, have demonstrated that the symbiosis signalling pathway is essential for a variety of symbiotic interactions: arbuscular mycorrhizal associations, rhizobial associations and Frankia associations. In arbuscular mycorrhizal and rhizobial associations this effect is absolute, these interactions are completely abolished in plants mutated in this signalling pathway. In contrast, the literature on a role for this signalling pathway in pathogen and parasite interactions is equivocal at best. Where effects have been reported they are generally subtle and do not compare to the absolute effects these genes have in symbiosis. There are studies in stable mutants and using RNAi and we feel that RNAi studies are a poor assessment for the function of this pathway, since off-target effects of the constructs cannot be ruled out. Below is the list of papers, we are aware, where stable mutants in the symbiosis pathway have been studied for effects with pests and pathogens:

1. Weerasinghe *et al.* (2005) reported that *NFR1* and *NFR5* were required for nematode infection, while *symrk* had a variable effect, a quote from this paper: “The *NFR1* and *NFR5* receptors can modulate the ability of root knot nematodes to establish feeding sites, although the role for *SYMRK* is less clear.”
2. Marcel *et al* (2010) showed that *castor* and *ccamk* mutants of rice were unaffected in colonisation by *Magnaporthe oryzae*.
3. Fernandez-Aparacio *et al.* (2010) showed that infection by the parasitic plant *Orobancha crenata* was increased in *dmi2* and *dmi3*.

4. Ben *et al.* (2013) showed that *dmi1*, but not *dmi2* or *dmi3*, had slightly increased resistance to the root fungal pathogen *Verticillium* wilt.
5. Rey *et al.* (2015) showed *dmi1*, *dmi2* and *dmi3* are unaffected in colonisation by *Phytophthora palmivora*.
6. Huisman *et al.* (2015) showed *dmi1*, *dmi2* and *dmi3* are unaffected in colonisation by *Phytophthora palmivora*.
7. Ried *et al.* (2019) reported that the *Arabidopsis* orthologs of *POLLUX* and two *NUPs* are necessary for the full reproductive success of *Hyaloperonospora arabidopsidis*, but have no role during colonisation of *E. cruciferarum* or *P. syringae*.

The majority of reports looking at the symbiosis signalling pathway during pathogen and pest colonisation have concluded this pathway has no effect on these interactions. The report on nematodes states that the results for *SYMRK* were equivocal. The results on *Verticillium* were limited to *DMII*, indicating a specific role for this gene, rather than the whole pathway. Thus the only report that really raises the possibility of pathway involvement is the work on *Orobanche* in *Medicago* and the vestigial members of this pathway in *Arabidopsis*, however, here effects were only observed for *Hpa*, but not the fungal or bacterial pathogens analysed. Considering that this pathway promotes intracellular infection by a variety of microorganisms and has been present in the plant kingdom for over 450 million years, it is surprising to us that there are not more examples where pests and pathogens have used this pathway, through the action of effectors, to drive their own colonisation. Instead a function for symbiosis signalling during colonisation by pathogens/parasites is very much the exception rather than the rule. There are surprisingly few examples that definitively demonstrate utilisation of the symbiosis signalling pathway during pest or pathogen interactions. From both our work (Bozsoki *et al* 2017 and in the current manuscript) and the work of Reid *et al* 2019, it is very clear that symbiosis signalling has no role in immunity: “Analysis of the multiplication of extracellular bacterial pathogens, *Hpa*-induced cell death or callose accumulation, as well as *Hpa*- or flg22-induced defence marker gene expression, did not reveal any traces of constitutive or exacerbated defence responses.” (Reid *et al* 2019)

We have now included our own analysis of a pathogenic interaction: *P. palmivora* and similar to previous reports see no evidence for symbiosis signalling during colonisation by this pathogen (Fig.6). We have also rewritten the discussion stating: “Symbiosis signalling has been present throughout much of plant evolution and since this signalling pathway promotes intracellular colonisation by microorganisms it seems a logical target for the action of effectors from pathogenic microorganisms. Surprisingly, little evidence for this yet exists, although mutants in homologs or orthologs of *Arabidopsis* symbiosis signalling genes show reduced fitness in colonising *Hyaloperonospora arabidopsidis*⁶⁷ and *dmi*, *dmi3* mutants of *Medicago* show enhanced colonisation by the parasitic plant *Orobanche*⁶⁸. In contrast, plants mutated in symbiosis signalling showed no effect during colonisation of *Arabidopsis* by *Pseudomonas syringae* or *Erysiphe cruciferarum*, colonisation of rice by *Magnaporthe grisea* or colonisation of *Medicago* by *Phytophthora palmivora*^{47-49,67}. The role of symbiosis signalling during symbiotic associations is absolute: mutations in this pathway completely abolish colonisation of arbuscular mycorrhizal fungi and rhizobial bacteria. No such major role has yet been found for this pathway during pathogen or parasitic associations, instead this pathway appears to play marginal roles in colonisation by a very limited set of

pathogens/parasites. A function for symbiosis signalling during pathogen/parasitic interactions is not the result of symbiosis signalling functioning during immunity^{7,67}, rather we suggest that some pathogens and parasites may manipulate this pathway for their own benefit.”

2) That the application of purified molecules can mimic the plant-directed signals generated by each symbiont. This aspect is particularly critical, because the authors are aware of the synergistic and complex chemical message that a mix of several different molecules may generate. Consequently, the dissection of plant responses based on single molecule application is a risky strategy that can be reasonable to support direct conclusions (e.g. CO8 triggers nuclear calcium spiking) but may have limited biological meaning if the conclusions extend too far from the experimental result (e.g. CO8 have or not a role in symbiosis). What evidence do we have that the application of one purified molecule to root cells that are growing under partial nutrient starvation is activating one ligand-specific pathway and not triggering a crosstalk with other pathways that start from structurally very similar LysM receptors? In fact, the matter of receptor complex composition is one huge open question in this field and trying to draw major conclusions in the absence of this piece of knowledge is, in my opinion, largely speculative.

Response: We feel the reviewer is being particularly unfair in the assessment of our work. Infact we are actually highlighting the importance of the mix of signals produced by the fungus, this features in the title, the abstract and two of the main figures. We end our manuscript stating: “Our work reveals that plants choose to encourage or restrict fungal colonisation, not based on the recognition of single signalling molecules, but rather through an integration of the mix of signalling molecules perceived”.

We cannot test mixes of signals that we are unaware of, so we have focused the study on those molecules that are strong candidates for mycorrhizal fungal signals. Exactly along the lines of this reviewers’ statement we have said in the discussion: “Hence, multiple receptors may contribute to mycorrhizal recognition of COs, LCOs and possibly other signalling molecules”. We began with the analysis of the signals in isolation and then looked at their effect in combination. We believe it would have been inappropriate to study these signals always in combination and we would have been equally criticised if we had not analysed these signals individually. We have also used germinated spore exudates, where appropriate. Science is reductionist by nature and we have to take a reductionist approach to understand the complexity of the system.

We have now added considerably more data on the effect of the mix of signals, looking at the timing, the genetic requirements and the need for protein synthesis to see an appropriate effect on the mix of signals.

3) That AM fungi produce CO8. I find particularly puzzling the importance given to CO8, a class of chitin oligomers that has not been characterized so far in AM fungal exudates and whose presence is depicted by the authors as a ‘very likely [...] byproduct of chitinaceous fungal cell walls’ (page 5). In fact, it is reasonable to speculate that chitin-related fungal signaling molecules derive from direct biosynthetic pathways rather than the dismantling of the crystalline chitin fibers in the wall (which is quite unlikely, in my opinion). Whatever their metabolic or catabolic origin, it remains to be demonstrated that this class of COs is produced by AM fungi during their interaction with the host plant.

Response: reviewer 1 states: “I agree that AM fungi are likely to produce CO8 (the alternative seems far more unlikely)”. Hence, this view of the reviewer is not shared by others in the field. We have shown that germinated spore exudates from AM fungi activate ROS in *Medicago* and *Arabidopsis* in a manner dependent on *CERK1*. This shows that AM fungi do produce elicitors that can be recognised by this receptor to activate immunity signalling. CO8 is a very strong candidate for such an elicitor as it is the only known component from fungi recognised by CERK1. We have now added into the text a statement that chitinaceous elicitors may be intentionally produced by AM fungi and their promotion by strigolactone treatment (Genre *et al.*, 2013) is in support of this. Such a model supports our observations that all chitinaceous molecules larger than CO4 promote symbiosis signalling.

Specific comments:

Page 3. The second paragraph states that “CO4 can activate immunity signalling”. The last paragraph in the same page reads “the plant actively suppresses immunity and this occurs following recognition of both CO4 and LCOs”. Is this not contradictory?

Response: This is a contradiction, but this contradiction exists in the literature. We have added the following sentence to the discussion: “Promotion of symbiosis signaling^{17, 20} and immunity signaling^{7, 8} and the suppression of immunity signaling³⁶ by CO4 appears contradictory, but it may be that the nature of the response is conditional on the status of the plant at the time of elicitation”.

The role of LCOs as shifters of the plant response toward symbiosis appears difficult to accept in the light of the normal AM phenotype of *nfp* mutants. The proposal that other LysM receptors may replace the *nfp* function remains to be demonstrated and falls within the abovementioned lack of information on the composition of receptor complexes. On the same line, the conclusion that LCOs ‘play an essential role in directing a symbiotic outcome’ (page 11) is in contrast with the observations in rice, where LCOs do not appear to be perceived by the root. My feeling is that the use of legumes is of limited help for the search of AM-specific signaling processes, due to the ability of these plants to interact with both fungal and bacterial symbionts. Overlaps and crosswalks between the respective signaling pathways cannot be excluded (and are also justified by their evolutionary relationship), which becomes especially critical when the roots are challenged with purified molecules (see my previous comment).

Response: We agree that legumes are challenging because of the dual symbioses these species can form. However, the model legumes have a wealth of genetics that makes these species amenable to study and they do form arbuscular mycorrhizal associations. The rice results eluded to are our own results and we have now found conditions for growing cereals that promote LCO recognition. When we have completed this body of work we will be preparing it for publication.

We agree that the system may be more complex than simply a CO receptor and an LCO receptor. We have eluded to this in the text particularly with regard to the function of *NFP*: “In addition to the role of *NFP* in LCO responses²⁰, we have observed some dependence on *NFP* in the CO responses. This is particularly apparent in the CO4 transcriptional response and the CO4 induction of calcium oscillations²⁰, but we also observed some dependence on *NFP* in the CO8 transcriptional response. Hence, *NFP* may be involved in the recognition of mycorrhizal-produced LCOs¹⁸, a role analogous to its function

during rhizobial associations, or alternatively, *NFP* may act in concert with other receptors during recognition of COs”.

The tests of calcium spiking activation in trichoblasts, presented in figure 1, is of limited interest in the frame of AM interactions, where this cell type is not involved in colonization. Other figures also consider atrichoblasts. I wonder why this important initial test was not done on atrichoblasts too. My personal interpretation of the graphs in fig 1 is that trichoblasts are more responsive to nod factor-like molecules than COs, which is perfectly in line with the depicted role of trichoblasts and atrichoblasts as targets of rhizobia and AM fungi, respectively.

Response: We HAVE shown calcium spiking in both trichoblasts and atrichoblasts in the manuscript. *Medicago* roots do have much greater sensitivity to *SmLCO* as depicted in Fig.1.

The comparison of gene expression patterns reveal a limited overlap between gene expression patterns induced by CO8 and CO4, while CO4 do activate more symbiosis- than defense-related genes compared to CO8. Taking into account my critic on the use of single molecules - or rather on the interpretation and biological significance of the results obtained from single molecule treatments – and our lack of knowledge about the composition of the receptor complex (es) that regulate such responses, I find it difficult to agree with the author conclusions.

Response: The reviewer is not correct about this interpretation of our transcriptome data: 73% of CO4 induced genes are also induced by CO8 demonstrating a very significant overlap between CO4 and CO8 induced genes. 61% of CO4-induced genes overlap with *flg22*, compared with 54% of CO8-induced genes that overlap with *flg22*. We do not understand what the reviewer is referring too in this statement, nor how they conclude that CO4 activates a greater symbiotic response. Based on the overlap with *flg22* we would argue that the majority of CO4 induced genes are immunity related.

We are dealing with a phosphate starved plant that is exposed to structurally very similar chitin-based molecules, which can be bound by several receptors with high structural similarity and variable affinities for each of these ligands. I would assume that under such conditions, whatever chitin-backbone molecule may at least partially activate non-specific pathways alongside its own, leading to vast overlaps in the resulting gene expression patterns. For this reason I would put the focus on differences (which may reveal specificities) rather than similarities between the expression patterns.

Response: Much of the CO8 induced genes that differ to LCO overlap with *flg22*. We have now added new data showing that these genes are activated by CO8 dependent on *CERK1* and *LYR4*, but independent of *DMI1* and *DMI2* (Fig.4 and Supplementary Figure.6).

The last paragraph of page 10 strongly points at the role of LCOs as true symbiotic signals, but what about rice, where LCOs are not active? As I mentioned above, I have the impression that studying these responses in legumes is not ideal, at least until we have clarified in depth what receptors rule AM interactions in plants (like rice) where there is no possible confusion with nod factor signalling.

Response: We agree that studying these responses in legumes is complicated by the dual symbioses these plants can form. We are currently undertaking work in cereals and analysing all of these signalling molecules in cereals. We see differences relative to what we reported in Sun *et al.* 2015, when we analysed mature cereal plants. I have presented this data at multiple conferences and I suspect this reviewer is aware of the data that we have. We are completing this body of work in cereals in preparation for a new submission on symbiotic responses in cereals.

The use of arabidopsis leaf treatments with *R. irregularis* exudate appears to me like a very unnatural condition. I can hardly understand what conclusions can be drawn from such an experiment, where a non host organ from a non host plant is exposed to the molecules released by a root-interacting symbiotic fungus.

Response: It is extremely well documented that *Arabidopsis* leaves respond to chitinaceous elicitors with activation of ROS and this is dependent on *CERK1*. The assay using GSE was to test whether we could see evidence for a similar chitinaceous elicitor in GSE. We find that GSE can induce ROS in *Arabidopsis* leaves and this response is greatly attenuated in the *cerk1* mutant. We conclude from this that an elicitor of *Arabidopsis* CERK1 is present in GSEs. We believe this elicitor is CO8, but we have been very cautious in this interpretation in the text. We believe this is a useful addition, but we have moved the data to supplementary.

In the same paragraph, DMI2 is presented as a critical member of the CSSP for the induction of symbiosis-specific responses. Nevertheless, Fig. 3 shows that the spiking response induced by CO8 and PGN is absent in *dmi2* mutants. Does this not indicate a role for DMI2 in defense? And consequently, does this not suggest that nuclear calcium spiking may also be present in the response to pathogens (also see my previous remark)? In fact this concept contrasts with figure 7d, where DMI2 is represented to have roles in both symbiotic and defense signalling

Response: As outlined in the first response to this reviewer, there is very little evidence for DMI2 functioning in plant defences and very much evidence to show that it functions in symbiosis. We believe that the much more logical interpretation is that CO8 and PGN activate symbiosis signalling. We have demonstrated this with calcium, with genetics and with gene expression. Reviewer 1 highlights how interesting and important the DMI2 result is and requested us to feature this result more strongly in the manuscript. We have modified the model in Fig.7 to clarify the role of DMI2.

Page 10, second paragraph, states that “What does appear to differentiate symbionts from pathogens is the presence of LCOs”. Has LCO production been demonstrated for all phylogenetic groups of AM fungi? And in what quantity compared to COs and compared to the concentrations used in the experiments? Again, how about non-legumes like rice, where LCOs are inactive?

Response: LCOs have been reported from multiple species of AM fungi and from many species of rhizobia. We don't believe that every single AM fungus must be analysed before the statement that symbionts produce LCOs is correct. As stated earlier our previous work in rice is not reflective of our current knowledge.

Figure 2. Why is panel d in this figure? It does not appear to have any relation with the figure title.

Response: This panel has been moved to supplementary material

Figure 4. Panel c should have a PGN label for uniformity with the remaining panels

Response: Changed as suggested.

Figure 7. Panel d: As mentioned above, the role of DMI2 does not correspond to what is stated in the text (and in suppl. Figure 2). Furthermore, where does CO4 position in this model?

Response: Model has been changed.

Suppl. Fig. 1. Panel c does not appear to belong to this figure (no connection with the figure title)

Response: Changed as suggested.

Suppl. Fig. 2." DMI2 is not required for CO8-induced defense responses". Then why is DMI2 in the scheme of figure 7d?

Response: Model has been changed.

Suppl. Fig. 3. Why did the authors choose to show the spiking patterns of the few PGN treated cells in panel e, when they did not for other treatments?

Response: We chose to present the majority situation, very few cells respond to PGN with calcium oscillations in *Mtcerk1* and *lyr4*. However, occasionally we can see responsive cells and we have provided traces to document this.

Suppl. Fig 5. I see important differences in the pattern of defense-related gene regulation between WTCO4 and WTCO8 (see my comment above).

Response: There are differences between CO4 and CO8, but as described above there are many more similarities between these elicitors than there are differences.

REVIEWERS' COMMENTS:

Reviewer #1 (Remarks to the Author):

This MS is vastly improved, with the new data on CO8/LCO antagonism and synergism really adding to its interest, and setting up intriguing future experiments with receptor complexes. Although it is very tightly written, I have a few comments below on some statements that require rewording. I thank the authors for their engaged responses to my previous review.

L107 "The CO8-induced nuclear calcium responses of roots in whole plants of *M. truncatula* 108 show a comparable periodicity and structure as those induced by CO4 (Fig. 1a)." This sentence is not a reasonable summation of the data, it makes it sound like the charted lines are much more highly correlated than they are - for example, the peaks don't overlap. Please tone the statement down or make it more accurate somehow.

L218 "Genes not induced by flg22 show extensive overlap between CO8 and SmLCO treatments 219 and mycorrhizal colonization (Fig. 4a)." I could not understand this sentence, please rewrite.

L251 "DMI2 is not required for plant 252 immunity (Supplementary Figure 4), suggesting that the differential activation of immunity 253 or symbiosis signaling by MtCERK1/LYR4 could be explained by the presence of DMI2 as a 254 co-receptor." I suggest changing "could" to "might".

Reviewer #3 (Remarks to the Author):

Feng et al., look into the role of COs and LCOs, and the receptors MtCERK1, LYR4, and NFP, for their involvement in initiating symbiosis and immunity signaling. The interesting angle to this work is that the authors show compelling data that long-chain COs (particularly CO8) trigger symbiosis signaling, whereas previously these were thought to be specific to triggering an immune response, and conversely that short-chain COs (particularly CO4) to initiate an immune response, whereas previously these were thought to be specific to symbiosis signaling. MtCERK1 and (to a lesser extent) LYR4 are shown to bind CO8 and CO4 (and peptidoglycan) and to be required for conferring downstream gene expression responses to these molecules. NFP is also highlighted as playing a role in down-regulating immunity in response to LCO, possibly in concert with a yet to be identified partner.

A strength of this work is that it thoroughly examines the roles of these chitinaceous molecules in initiating signaling in the plant host, both immune- and symbiosis-related, as well as the involvement of MtCERK1 and LYR4 as receptors for CO8 and CO4 (and peptidoglycan). In doing so, this manuscript brings together various observations reported over many years (eg, Maillet et al, 2011; Genre et al., 2013) into a cohesive model. I find their conclusions (eg, that a mix of COs and LCOs shifts towards a symbiotic, and not immunogenic, response) to be generally well supported by the data in the manuscript.

A few points that I think worth commenting on, for the author's consideration.

About the issue of whether it is truly CO8 initiating the symbiosis signaling - Yes, treatment of acetazolamide with CO8 partially protects this molecule from plant chitinases, and the authors show that co-treatment of this chemical with CO8 also increases the host's symbiotic and immunologic responses, possibly (as the authors interpret) because of the increased concentration of CO8 available to the plants.

However, it is clear that more than 40% of CO8 has been converted to something else (most likely

a mix of shorter chain COs) within minutes of exposure to roots, even in the acetazolamide treated samples. Thus, 'CO8' samples are no longer purely CO8 after exposure to roots, but are most likely a mix of CO8 + shorter COs, and it seems reasonable (to me) to assume that acetazolamide treatment would equally protect any shorter chain COs that might result from CO8 degradation. By this logic, it is formally possible that the enhanced plant responsiveness (both symbiotic and/or immunologic) observed in CO8 plus acetazolamide samples is due to the increased stability of these short chain COs, and not due to the increased concentration of CO8. And the time frames required for the assays (ex, calcium signaling assays, ROS, etc) are more than sufficient to allow degradation of CO8 into a mix of shorter COs for all these experiments. For this reason, the authors statement that "We conclude that CO8 itself must act as the elicitor...and CO8 elicitation cannot be explained by breakdown products..." is a bit strong. Moreover, this limitation (ie, the possibility that all "CO8" treatments become, in reality, "CO8 plus a mix of shorter COs") will extend to all experiments in which CO8 is applied to roots (eg, including the gene expression data). What is necessary to really address this is a Medicago mutant that does not produce any of these chitinases...yet that is a bit much to ask. Moreover, these data must be interpreted within the context of the whole manuscript, which forms a consistent argument that CO8 has a role relevant to symbiosis signaling - data showing the binding of MtCERK1 (and LYR4) to CO8 beads, and the demonstrated involvement of these receptors in mediating symbiosis signaling - is particularly compelling in support of the authors overall conclusions regarding CO8 and symbiosis. Thus, while I am of the opinion that the authors cannot definitely rule out the possibility of shorter chain COs (and not CO8 itself) initiating the symbiosis signaling in experiments involving roots, I nonetheless agree that their interpretation is reasonable given the data presented as a whole. As regards comments provided by previous reviewers - in particular reviewer 2 - I think the authors statement that AMF likely produce CO8 is reasonable (and is consistent with the ROS data in Sup Fig 3).

I am more concerned with the use of SmLCO as a proxy for LCO produced by AMF, based on the authors statement that S-LCO and SmLCO show 'similar elicitation activity...for induction of symbiosis signaling in *M. truncatula*'. I find this statement misleading, as regards gene expression. Whereas Nod-factor from *S. meliloti* and Myc-LCOs were found (in Sun et al, 2015) to trigger similar Ca²⁺-signaling oscillations, the gene expression response of Medicago was found to differ between these LCOs. I'm surprised that the authors did not use NS-LCO for their gene expression studies (since they do have access to this - Fig 1) in lieu of SmLCO, and suggest the authors modify (or qualify) their statement suggesting that SmLCO and Myc-LCOs generate comparable output, within the overall context of gene expression.

I'm also intrigued by apparent differences in expression profiles elicited by CO4 and CO8 in wt plants - the authors simply state that 'The degree of transcriptional response to CO4 was greatly attenuated compared to CO8...' which is consistent with the rest of their data...but to my eyes, the heat maps look more dissimilar than I would expect from a mere decrease in responsiveness, and I note a group of genes that are specifically up-regulated by CO4 and not CO8. In stating this, I acknowledge that I have not thoroughly assessed the gene expression data and my comparison of heat maps may simply be inaccurate...however, I also note that reviewer 2 raised the issue of dissimilarity between CO4 and CO8 expression data, so I do think this is an opportunity for the authors to add a bit more text about this data to interpret for readers.

Line 321 - correct spelling of atrichoblasts

Line 345 - as phrased, it is unclear whether the authors mean reduced Hpa fitness, or reduced Arabidopsis fitness during Hpa infection

Lines 338-339 - I find this paragraph a bit awkward the way it is written but the information itself is very relevant and important to discuss. For example, the statement that plant species which have lost AM associations also lose the sym signaling pathway appears to be contradictory to statements that immediately follow referring to the signaling pathway in Arabidopsis (which

shouldn't have this pathway, based on the statement in lines 338-339, as it doesn't engage in AM symbiosis). Having now read response to reviewer 2, I realize this has most likely been added into the text in response to reviewer comments...but I had already suspected something like this to be the case from my first reading of the manuscript – it just felt disjointed. I'd encourage the authors to rework this text slightly.

REVIEWERS' COMMENTS:

Reviewer #1 (Remarks to the Author):

This MS is vastly improved, with the new data on CO8/LCO antagonism and synergism really adding to its interest, and setting up intriguing future experiments with receptor complexes. Although it is very tightly written, I have a few comments below on some statements that require rewording. I thank the authors for their engaged responses to my previous review.

L107 "The CO8-induced nuclear calcium responses of roots in whole plants of *M. truncatula* 108 show a comparable periodicity and structure as those induced by CO4 (Fig. 1a)." This sentence is not a reasonable summation of the data, it makes it sound like the charted lines are much more highly correlated than they are - for example, the peaks don't overlap. Please tone the statement down or make it more accurate somehow.

Response: We have analysed many different traces of CO4 and CO8 induced calcium oscillations. While individual traces can differ, we do not believe that there is a relevant difference in the overall nature of the responses induced by these two elicitors. In an attempt to clarify this intent we have written: "The CO8-induced nuclear calcium responses of roots in whole plants of *M. truncatula* show a periodicity similar in nature to those induced by CO4"

L218 "Genes not induced by flg22 show extensive overlap between CO8 and SmLCO treatments 219 and mycorrhizal colonization (Fig. 4a)." I could not understand this sentence, please rewrite.

Response: We agree this was a clumsy sentence. We have rewritten to "A second group of genes are induced by CO8, *SmLCO* and mycorrhizal colonization (Fig. 4a), but not flg22."

L251 "DMI2 is not required for plant 252 immunity (Supplementary Figure 4), suggesting that the differential activation of immunity 253 or symbiosis signaling by MtCERK1/LYR4 could be explained by the presence of DMI2 as a 254 co-receptor." I suggest changing "could" to "might".

Response: We made the change as suggested.

Reviewer #3 (Remarks to the Author):

Feng et al., look into the role of COs and LCOs, and the receptors MtCERK1, LYR4, and NFP, for their involvement in initiating symbiosis and immunity signaling. The interesting angle to this work is that the authors show compelling data that long-chain COs (particularly CO8) trigger symbiosis signaling, whereas previously these were thought to be specific to triggering an immune response, and conversely that short-chain COs (particularly CO4) to initiate an immune response, whereas previously these were thought to be specific to symbiosis signaling. MtCERK1 and (to a lesser extent) LYR4 are shown to bind CO8 and CO4 (and peptidoglycan) and to be required for conferring downstream gene expression responses to these molecules. NFP is also highlighted as playing a role in down-regulating immunity in response to LCO, possibly in concert with a yet to be identified partner.

A strength of this work is that it thoroughly examines the roles of these chitinaceous molecules in

initiating signaling in the plant host, both immune- and symbiosis-related, as well as the involvement of MtCERK1 and LYR4 as receptors for CO8 and CO4 (and peptidoglycan). In doing so, this manuscript brings together various observations reported over many years (eg, Maillet et al, 2011; Genre et al., 2013) into a cohesive model. I find their conclusions (eg, that a mix of COs and LCOs shifts towards a symbiotic, and not immunogenic, response) to be generally well supported by the data in the manuscript.

A few points that I think worth commenting on, for the author's consideration.

About the issue of whether it is truly CO8 initiating the symbiosis signaling - Yes, treatment of acetazolamide with CO8 partially protects this molecule from plant chitinases, and the authors show that co-treatment of this chemical with CO8 also increases the host's symbiotic and immunologic responses, possibly (as the authors interpret) because of the increased concentration of CO8 available to the plants.

However, it is clear that more than 40% of CO8 has been converted to something else (most likely a mix of shorter chain COs) within minutes of exposure to roots, even in the acetazolamide treated samples. Thus, 'CO8' samples are no longer purely CO8 after exposure to roots, but are most likely a mix of CO8 + shorter COs, and it seems reasonable (to me) to assume that acetazolamide treatment would equally protect any shorter chain COs that might result from CO8 degradation. By this logic, it is formally possible that the enhanced plant responsiveness (both symbiotic and/or immunologic) observed in CO8 plus acetazolamide samples is due to the increased stability of these short chain COs, and not due to the increased concentration of CO8. And the time frames required for the assays (ex, calcium signaling assays, ROS, etc) are more than sufficient to allow degradation of CO8 into a mix of shorter COs for all these experiments. For this reason, the authors statement that "We conclude that CO8 itself must act as the elicitor...and CO8 elicitation cannot be explained by breakdown products..." is a bit strong. Moreover, this limitation (ie, the possibility that all "CO8" treatments become, in reality, "CO8 plus a mix of shorter COs") will extend to all experiments in which CO8 is applied to roots (eg, including the gene expression data). What is necessary to really address this is a *Medicago* mutant that does not produce any of these chitinases...yet that is a bit much to ask. Moreover, these data must be interpreted within the context of the whole manuscript, which forms a consistent argument that CO8 has a role relevant to symbiosis signaling – data showing the binding of MtCERK1 (and LYR4) to CO8 beads, and the demonstrated involvement of these receptors in mediating symbiosis signaling - is particularly compelling in support of the authors overall conclusions regarding CO8 and symbiosis. Thus, while I am of the opinion that the authors cannot definitely rule out the possibility of shorter chain COs (and not CO8 itself) initiating the symbiosis signaling in experiments involving roots, I nonetheless agree that their interpretation is reasonable given the data presented as a whole. As regards comments provided by previous reviewers – in particular reviewer 2 – I think the authors statement that AMF likely produce CO8 is reasonable (and is consistent with the ROS data in Sup Fig 3).

Response: We appreciate the reviewer's feedback and agree that the best experiment to address this point would be to check CO8-induced calcium responses in *Medicago* chitinase mutants. However, the highly redundant nature of chitinases makes this a very challenging approach. We believe that the combination of genetic, pharmacological and biochemical data, all point at CO8 acting as an elicitor for symbiosis signalling. However, we have now added the following statement acknowledging that breakdown products of CO8 could also be elicitors: "We conclude that CO8 itself most likely acts as the elicitor, but breakdown products of CO8 can also act as elicitors, provided they are larger than CO3."

I am more concerned with the use of SmLCO as a proxy for LCO produced by AMF, based on the authors statement that S-LCO and SmLCO show 'similar elicitation activity...for induction of symbiosis signaling in *M. truncatula*'. I find this statement misleading, as regards gene expression. Whereas Nod-factor from *S. meliloti* and Myc-LCOs were found (in Sun et al, 2015) to trigger similar Ca²⁺-signaling oscillations, the gene expression response of Medicago was found to differ between these LCOs. I'm surprised that the authors did not use NS-LCO for their gene expression studies (since they do have access to this – Fig 1) in lieu of SmLCO, and suggest the authors modify (or qualify) their statement suggesting that *SmLCO* and Myc-LCOs generate comparable output, within the overall context of gene expression.

Response: We previously demonstrated that all LCOs induce calcium spiking and this is *NFP* dependent, regardless of the modifications to LCOs. For the RNA-seq analysis, we were interested in the degree of overlap between gene induction resulting from CO and LCO treatments. We find a very significant overlap between CO8 and *SmLCO* in their gene expression profiles, showing that the activation of symbiosis signalling by these elicitors gives similar gene expression changes. Different LCOs may give subtly different outputs, but we believe that any such differences would not change our conclusions: that LCO and CO8 induced gene expression changes are highly overlapping.

I'm also intrigued by apparent differences in expression profiles elicited by CO4 and CO8 in wt plants – the authors simply state that 'The degree of transcriptional response to CO4 was greatly attenuated compared to CO8...' which is consistent with the rest of their data...but to my eyes, the heat maps look more dissimilar than I would expect from a mere decrease in responsiveness, and I note a group of genes that are specifically up-regulated by CO4 and not CO8. In stating this, I acknowledge that I have not thoroughly assessed the gene expression data and my comparison of heat maps may simply be inaccurate...however, I also note that reviewer 2 raised the issue of dissimilarity between CO4 and CO8 expression data, so I do think this is an opportunity for the authors to add a bit more text about this data to interpret for readers.

Response: We appreciate the reviewer highlighting this point and have added a statement that some of the CO4 response appears to be specific.

Line 321 – correct spelling of atrichoblasts

Response: Corrected

Line 345 – as phrased, it is unclear whether the authors mean reduced Hpa fitness, or reduced Arabidopsis fitness during Hpa infection

Response: We modified the text.

Lines 338-339 – I find this paragraph a bit awkward the way it is written but the information itself is very relevant and important to discuss. For example, the statement that plant species which have lost AM associations also lose the sym signaling pathway appears to be contradictory to statements that immediately follow referring to the signaling pathway in Arabidopsis (which shouldn't have this pathway, based on the statement in lines 338-339, as it doesn't engage in AM symbiosis). Having now read response to reviewer 2, I realize this has most likely been added into the text in response to reviewer comments...but I had already suspected something like this to be the case from my first

reading of the manuscript – it just felt disjointed. I'd encourage the authors to rework this text slightly.

Response: The reviewer is correct. We were forced to extend considerably this argument by the previous reviewer 2 and we agree this was not beneficial to the clarity of the manuscript. We have now shortened this text for clarity, without, in our opinion, removing any of the points that were made. We hope the new paragraph is now clearer regarding the points that we raise.